# Zeroth-Order Adaptive Neuron Alignment Based Pruning without Re-Training

**Elia Cunegatti**                                                          *elia.cunegatti@unitn.it*
*University of Trento, Italy*

**Leonardo Lucio Custode**                                          *leonardo.custode@gmail.com*
*Independent Researcher*

**Giovanni Iacca**                                                       *giovanni.iacca@unitn.com*
*University of Trento, Italy*

**Reviewed on OpenReview:** *https://openreview.net/forum?id=uPyNaNqFK2*

## Abstract

Network pruning focuses on algorithms that aim to reduce a given model's computational cost by removing a subset of its parameters while having minimal impact on performance. Throughout the last decade, the most widely used pruning paradigm has been pruning and re-training, which nowadays is inconvenient due to the vast amount of pre-trained models, which are, in any case, too expensive to re-train. In this paper, we exploit functional information from dense pre-trained models, i.e., their input activations, to obtain sparse models that maximize the activations' alignment with respect to their corresponding dense models. Hence, we propose NEURONAL, a *top-up* algorithm that can be used on top of any given pruning algorithm for LLMs, which modifies the block-wise and row-wise sparsity, exploiting information from both the dense model and its sparse version to maximize the *neuron alignment* among activations. Different from existing methods, our approach adaptively selects the best hyperparameters for the block-wise and row-wise sparsity ratios w.r.t. the model and the desired sparsity, and requires *no re-training*. We test our method over ∼300 test cases with four LLM families, three sparsity ratios, and ten language tasks (three language modeling and seven zero-shot datasets), showing how it consistently outperforms the latest state-of-the-art methods in terms of performance-runtime trade-off. The code is available at https://github.com/eliacunegatti/NeuroAL.

## 1 Introduction

In recent times, Large Language Models (LLMs) have shown incredible performance over several language tasks (Wei et al., 2022; Min et al., 2023; Chang et al., 2024). However, their performance usually improves with their sizes (i.e., the number of trainable parameters), which in turn is proportional to the computational cost of training and inference. One way to reduce this cost is *network pruning*, which studies algorithms that remove parameters while minimizing performance degradation. This approach, extensively studied on Convolutional Neural Networks (CNNs) (Frankle & Carbin, 2019; Lee et al., 2019; Wang et al., 2020; Evci et al., 2020), nowadays is mainly applied to pre-trained models (Touvron et al., 2023a;b; Jiang et al., 2023).

This shift has required a change of paradigm in pruning techniques: in fact, while in CNNs the main paradigm is iterative pruning (with re-training) (Frankle & Carbin, 2019), with pre-trained models (such as LLMs), it is not possible in most cases to perform a full re-training, because (1) training data are often not accessible, and (2) full re-training would be anyway too expensive. This calls for "exploiting" as much as possible the information contained in a pre-trained model to obtain a performant sparse version of it, using weights' information (Jaiswal et al., 2024), activations (Sun et al., 2023; 2024), or reconstruction error (Frantar & Alistarh, 2023), without the need for re-training. More recently, a new category of pruning algorithms, which we may call *top-up* algorithms (i.e., methods that can be applied on top of a given pruning algorithm for

LLMs), has emerged, aiming at further improving pruning performance. Such approaches can be divided into two categories: those that minimize the reconstruction error (Guo et al., 2024; Xu et al., 2024; Zhang et al., 2024), and those that impose non-uniform sparsity distribution, modifying the block-wise sparsity (Yin et al., 2024; Lu et al., 2024; Li et al., 2024). The latter category is extremely effective on CNNs (Frankle et al., 2020; Su et al., 2020), while its application to LLMs has only recently emerged.

**Contributions** In this paper, we first analyze the major limitations of current *top-up* algorithms. To do so, we carefully analyze the state-of-the-art top-up methods, highlighting their limitations in terms of sensitivity to hyperparameters, the required computational budget, and their block-importance metric. Leveraging this knowledge, we introduce NEURONAL, a novel *top-up* pruning algorithm that outperforms, in most cases, the previous state-of-the-art approaches over both Language Modeling datasets and Zero-Shot tasks, while providing hyperparameter adaptation and reduced runtime to obtain the non-uniform sparsity allocation.

The algorithm consists of a two-step approach that re-distributes the block-wise sparsity, i.e., the sparsity among Transformer blocks, and the row-wise sparsity, i.e., the sparsity for each row of a given layer's matrix, maximizing a metric which exploits information from both the dense and sparse model, namely the *neuron alignment* between dense and sparse activations. NEURONAL does not require the user to specify any hyperparameter-tuning, as it automatically selects the most-performing values from a suitable set, hence adapting to the underlying model and the target sparsity. Another advantage is that the *neuron alignment* only requires the computation of the activations of the dense and sparse models, which reduces the computation budget required, compared to other *top-up* approaches.

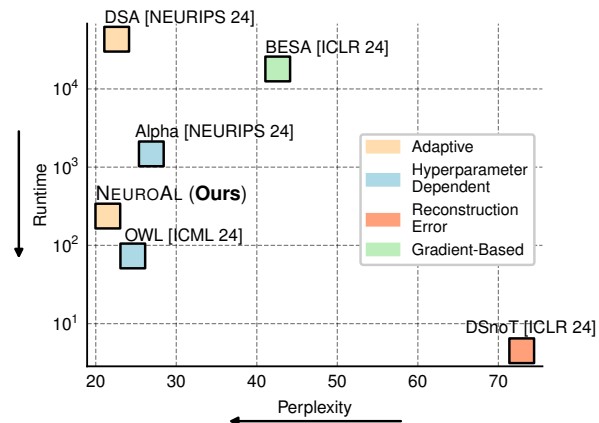

Figure 1: Perplexity vs. Runtime (seconds) trade-off among different *top-up* algorithms and our proposed NEURONAL based on LLama-1 7B with a sparsity of 70%, evaluated on WikiText2.

We test our approach on three Language Modeling datasets and seven Zero-Shot tasks over four different LLM families from 7B to 70B parameters, to show its ability to outperform, in the majority of the cases, the most recent state-of-the-art techniques, including OWL (Yin et al., 2024), DsNoT (Zhang et al., 2024), and AlphaPruning (Lu et al., 2024) over three different high sparsity values (60%, 70%, and 80%) for a total of 276 test-cases. The performance results clearly show that our proposed NEURONAL stands out as the most performing *top-up* pruning algorithm, outperforming in most cases all baselines, and on average by a large margin, over both Language Modeling and, especially, Zero-Shot cases. Furthermore, the runtime analysis indicates that our proposed approach outperforms the state-of-the-art baselines while also being more time efficient, being up to ~20× faster than the closest competitor over 70B models. To assess the robustness of our approach, we also conduct an in-depth sensitivity analysis.

## 2 Related Work

In this section, we provide a comprehensive discussion about network pruning applied to LLMs. We first introduce structured and unstructured network pruning; then, we focus on the latter, introducing the latest approaches proposed for improving sparse model performance.

### 2.1 Structured Network Pruning

Given a layer's weight matrix $W \in \mathbb{R}^{n \times m}$ to sparsify, structured pruning removes either entire rows ($n$) or columns ($m$) (see the next section), aiming at speeding up both training and inference time. The first approach that applies structured pruning to LLMs has been proposed by Ma et al. (2023), and focuses on the dependency of Transformers, i.e., it removes components of the networks while maximizing their original functionality. In (Kurtić et al., 2024), a pruning mechanism has been devised to remove components with the worst balance between loss and runtime. Other structured pruning approaches have been proposed based

on combinatorial optimization (Meng et al., 2024), perturbative forward-pass only (Dery et al., 2024), and reduction of the embedding dimension through PCA (Ashkboos et al., 2023). Finally, in (Gromov et al., 2024) it has been found that the last Transformer blocks are redundant, hence they can be completely removed with minor performance drops. The reason behind this phenomenon lies in the similarity between the learnable representation of consecutive blocks, which turns out to increase when the block depth increases. While all these approaches can achieve valuable inference speed-ups, the performance of the resulting sparse models w.r.t. their dense counterparts can be matched only at low sparsity values, such as 20% in (Ma et al., 2023) or 30% in (Ashkboos et al., 2023). This somehow limits the applicability of these methods, since in the case of models with billions of parameters, one may need more aggressive pruning strategies to meet stringent hardware requirements.

## 2.2 Unstructured Network Pruning

Differently from structure pruning, unstructured pruning works by removing weights in a scattered (i.e., non-structured) way. While in this setting the inference speed-up is limited (although techniques for reordering weights are available (Li et al., 2019; Mishra et al., 2021; Zhou et al., 2021)), the performance w.r.t. the dense model can be preserved also at high sparsity ratios (i.e., above 50%), with the performance at lower sparsity being almost always completely preserved. The first approach of this kind has been proposed in (Frantar & Alistarh, 2023), where weight pruning and reconstruction are combined based on the Hessian matrix. Even a simple magnitude-based approach turned out to perform well (Jaiswal et al., 2024), also when integrated with information on the neuron activations (Sun et al., 2023; Farina et al., 2024). These approaches compute a score for each weight and then remove the ones with the lower scores for each layer, with a uniform sparsity across layers.

## 2.3 Top-Up Algorithms

To improve the performance of unstructured pruning, several *top-up* algorithms have been devised. These approaches can be categorized into two distinct groups: methods that minimize the reconstruction error, keeping the sparsity uniform for each block, and methods that modify the block-wise sparsity of the model, resulting in non-uniform sparsity distribution across blocks.

The first group firstly sparsifies the model using a pruning algorithm and then, either dynamically (Zhang et al., 2024) or by backpropagation (Guo et al., 2024), updates the pruning mask. The second group (to which our method belongs) modifies the block-wise sparsity (obtained by a given pruning algorithm) based either on activations' outliers (Yin et al., 2024), Empirical Spectral Distance (ESD) (Lu et al., 2024), or allocation functions in a gradient-free manner (Li et al., 2024), while in BESA (Xu et al., 2024) gradient information is used to set layer-wise sparsity using block-wise reconstruction error.

The idea of simply redistributing the layer-wise sparsity is known to be extremely well-performing on Multi-Layer Perceptrons (MLPs) and Convolutional Neural Networks (CNNs). The first approach of this kind, based on the Erdős–Rényi (ER) model, has been proposed by Mocanu et al. (2018) for MLPs and then adjusted for CNNs in (Evci et al., 2020), while an empirical study about the effect of layer-wise pruning using different sparsity ratios has been done in (Liu et al., 2021). Regarding Transformers (both for vision and text), the state-of-the-art algorithms (Frantar & Alistarh, 2023; Sun et al., 2023) have been devised to set the block-wise sparsity across the Transformer blocks in a uniform way. Later on, OWL (Yin et al., 2024), AlphaPruning (Lu et al., 2024), and DSA (Li et al., 2024) have been proposed to build upon scoring-based pruning algorithms, adjusting the block-wise sparsity in a non-uniform way. These approaches improve the performance of several pruning algorithms, e.g. (Frantar & Alistarh, 2023; Sun et al., 2023), especially at sparsity above 60%. On the same line, BESA (Xu et al., 2024) allocates layer-wise sparsity across each block's layer using gradient information. Recently, modality-wise sparsity distribution has been investigated in the case of multimodal tasks in (Farina et al., 2024; He & Chen, 2024).

## 3 Current limitations of top-up algorithms

We discuss below the three main limitations of the state-of-the-art approaches for redistribution of non-uniform sparsity in LLMs, namely 1) their need for hyperparameter tuning, 2) their large runtime, and 3) their block importance metric calculation (hence their block sparsity allocation).

### 3.1 Need for Hyperparameter Tuning

We analyze the sensitivity of the hyperparameters used by OWL, namely $\lambda$ and $M$, and by AlphaPruning, namely $\epsilon$. Concerning OWL, the first hyperparameter is used to set how much the sparsity can vary across blocks (i.e., $[s - \lambda, s + \lambda]$) while keeping the overall sparsity fixed as $s$. The second hyperparameter, $M$, defines the outliers' threshold: namely, for each block, the number of outliers is computed as the number of activations that are $M$ times greater than the block's activations' mean. For AlphaPruning, instead, a hyperparameter called $\epsilon$ is used and manually tuned to set two tunable hyperparameters $(s_1, s_2)$ that control the sparsity across blocks. We test the sensitivity of OWL and AlphaPruning to their hyperparameters, using three different sparsity ratios, two LLMs, and Wanda as the underlying pruning algorithm. Fig. 2 displays the perplexity on WikiText2 of the different hyperparameter settings obtained with OWL (first two rows) and AlphaPruning (last row); the gray square corresponds to the best *a-posteriori* hyperparameter selection. It can be seen that no single hyperparameter value achieves the best performance in all settings, which entails that careful tuning is required for these approaches to be effective.

### 3.2 Large Runtime

Another main limitation of some of the current approaches for non-uniform distribution is their computational runtime. This holds mainly for BESA (Xu et al., 2024) and DSA (Li et al., 2024). On LLama-1 7B, the first approach, which relies on gradient information, requires ∼5 hours to find the best non-uniform distribution configuration. On the other hand, DSA uses an evolutionary approach to find the best combination of a set of allocation functions, requiring ∼12 hours to find the best distribution[1]. This high runtime is due to the evaluation of each sparse model obtained by applying all the possible combinations of sparsity allocation functions.

### 3.3 Block Importance Metric

Almost all the *top-up* algorithms select the blockwise sparsity w.r.t. the block importance given by scoring criteria computed either from the dense model or from the evaluation of the sparse model. In the first case, the block importance is computed on the dense models on a block-wise view using either outlier information, as in OWL, or the ESD, as in AlphaPruning, without focusing on how the selected non-uniform sparsity, once applied, could change the block importance of the successive layers.

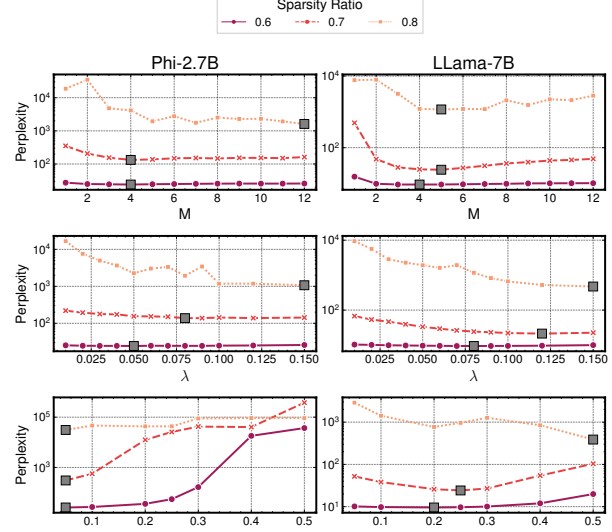

Figure 2: Perplexity for various hyperparameter settings of OWL ($M$,$\lambda$) and AlphaPruning ($\epsilon$) using Phi-2 and LLama-1 7B for three sparsity ratios. The gray square corresponds to the hyperparameter values that lead to the best performance.

On the other hand, DSA only uses the perplexity of the sparse model as the evaluation metric. The only exception is BESA, which, similarly to NEURONAL, uses information from both the dense and the sparse model. However, it does so using gradient information, which leads to a high computational runtime. Also, DSnoT minimizes the reconstruction error using information from both the dense and sparse models. However, it only updates each layer's binary mask without changing the sparsity distribution across blocks.

Table 1: Properties of state-of-the-art *top-up* algorithms vs. NEURONAL. The runtime is expressed as orders of magnitude computed on LLama-1 7B. For the metrics, we indicate whether they are computed over the dense ($\mathcal{D}$) and/or sparse ($\mathcal{S}$) models.

| Top-Up | Hyper. Tuning | Runtime (s) | Metric |
|---|---|---|---|
| DSnoT (Zhang et al., 2024) | ✓ | $10^0$ | Reconstruction Error ($\mathcal{D},\mathcal{S}$) |
| BESA (Xu et al., 2024) | ✗ | $10^4$ | Reconstruction Loss ($\mathcal{D},\mathcal{S}$) |
| OWL (Yin et al., 2024) | ✓ | $10^2$ | Outliers ($\mathcal{D}$) |
| AlphaPruning (Lu et al., 2024) | ✓ | $10^3$ | ESDs ($\mathcal{D}$) |
| DSA (Li et al., 2024) | ✗ | $10^4$ | Perplexity ($\mathcal{S}$) |
| NEURONAL (Ours) | ✗ | $10^2$ | Neuron Alignment ($\mathcal{D},\mathcal{S}$) |

---

[1]The runtime has been taken from the original papers, where authors used A100 GPUs for BESA, and H800 GPUs for DSA.

Table 1 summarizes the different properties of the current *top-up* algorithms and shows how our proposed approach positions w.r.t. the previously proposed approaches.

## 4    Methodology

We now describe our proposed NEURONAL.

**Preliminaries**. Given a dense model $\mathcal{D}$, a pruning algorithm $\mathcal{P}$, and a target sparsity $s \in [0, 1]$, unstructured pruning assigns a saliency score $\Psi$ to each weight $w \in \mathcal{D}$ and use a $\texttt{top}_s(\cdot)$ function to retain the $(1-s)|\mathcal{D}|$ weights with the highest values. This function returns a binary mask $\mathcal{M}$, such that the pruned model can be obtained as $\mathcal{S} = \mathcal{D} \odot \mathcal{M}$ (where $\odot$ represents the Hadamard product). In the case of LLMs, since these models are composed of stacked Transformer blocks (each one denoted as $\mathcal{B}_i$), i.e., sets of linear layers (each one defined by a weight matrix $\mathbf{W}^{\ell_i^j}$) that implement the self-attention mechanism followed by an MLP, the mask is usually computed with a *uniform* sparsity level across each layer $\ell_i^j$ (Frantar & Alistarh, 2023), as:

$$\mathcal{M}^{\ell_i^j} = \texttt{top}_{s^{\ell_i^j}}(\Psi^{\ell_i^j}, \mathbf{W}^{\ell_i^j}). \tag{1}$$

**Overview of NeuronAl**. Our proposed NEURONAL is based on two principles: (1) *neuron alignment*, which compares internal input activations rather than layer outputs as in (Frantar & Alistarh, 2023) (see Appendix A), and (ii) *adaptive sparsity reallocation*, which removes the need to manually tune the algorithm hyperparameters. NEURONAL relies on neuron alignment to reassign sparsity in two stages: firstly across blocks (via vectors $\mathbf{s}^{\mathcal{B}}$), and then across rows (via vectors $\mathbf{s}^r$). Both steps aim to minimize the discrepancy between the dense and sparse models' internal input activations, as formalized in Section 4.1. Importantly, the method requires no weight updates, gradients, or retraining. It simply reuses the scoring function $\Psi$ from the selected pruning method $\mathcal{P}$ and adjusts accordingly the sparsity ratios.

### 4.1    Neuron Alignment

The rationale behind NEURONAL is to adapt the sparsity distribution by comparing how well the sparse model preserves the internal representations of the dense one, a concept that we refer to as neuron alignment. Rather than measuring the discrepancy in layer outputs, as in (Frantar & Alistarh, 2023), our method compares the *input activations* to each projection matrix in a Transformer block.

Here, we formalize the proposed neuron alignment metric, which relies on the block and row-wise sparsity allocation. Given $\mathcal{D}$ and its sparse version $\mathcal{S} = \mathcal{D} \odot \mathcal{M}$, obtained via a pruning method $\mathcal{P}$ at sparsity ratio $s$, we firstly perform forward passes over a calibration set $C_\lambda$ and collect the activations $\mathcal{A}_{\mathcal{D}}$ and $\mathcal{A}_{\mathcal{S}}$. Then, for each Transformer block $\mathcal{B}_i$, we extract the input activations to all linear projection layers (i.e., the vectors $\mathbf{a}^{\ell_i^j}$ such that the corresponding layer computes $\mathbf{a}^{\ell_i^j}\mathbf{W}^{\ell_i^j}$ during the forward pass). This includes projections from the attention module ($\mathbf{W_Q}, \mathbf{W_K}, \mathbf{W_V}, \mathbf{W_O}$) and the MLP module ($\mathbf{W}_{\text{gate}}, \mathbf{W}_{\text{up}}, \mathbf{W}_{\text{down}}$)[2].

To make the comparison consistent across layers, we normalize each input activation vector as $\mathbf{a}^{\ell_i^j} / \sum \mathbf{a}^{\ell_i^j}$. For a given projection layer $\ell_i^j$, given an input $x \in C_\lambda$, we define its neuron alignment score as:

$$\text{neuron}_{\text{al}}(A_{\mathcal{D}}^{\ell_i^j}, A_{\mathcal{S}}^{\ell_i^j}, x) = \frac{1}{|A_{\mathcal{D}}^{\ell_i^j}(x)|} \left\| \frac{A_{\mathcal{D}}^{\ell_i^j}(x)}{\sum A_{\mathcal{D}}^{\ell_i^j}(x)} - \frac{A_{\mathcal{S}}^{\ell_i^j}(x)}{\sum A_{\mathcal{S}}^{\ell_i^j}(x)} \right\|_2 \tag{2}$$

and then the neuron alignment across the whole model as:

$$\texttt{NeuronAL}(\mathcal{D}, \mathcal{S}, C_\lambda) = \sum_{x \in C_\lambda} \sum_{\mathcal{B}_i} \sum_{\ell_i^j \in \mathcal{B}_i} \text{neuron}_{\text{al}}(A_{\mathcal{D}}^{\ell_i^j}, A_{\mathcal{S}}^{\ell_i^j}, x). \tag{3}$$

---

[2]The input activations $\mathbf{a}^{\ell_i^j}$ used for calculating the neuron alignment are either taken right after the LayerNorm (for $\mathbf{W_Q}$, $\mathbf{W_K}, \mathbf{W_V}, \mathbf{W}_{\text{gate}}$, and $\mathbf{W}_{\text{up}}$), or from the intermediate values within each sub-block (for $\mathbf{W_O}$, and $\mathbf{W}_{\text{down}}$).

### 4.1.1 Block-wise Sparsity Ratio

First, we optimize sparsity at the granularity of transformer blocks. Given a set of candidate sparsity window $\lambda^{\mathrm{set}}$, we generate multiple vectors $\mathbf{s}^{\mathcal{B}}_{set} = \{\mathbf{s}^{\mathcal{B}}_{\lambda_1}, \mathbf{s}^{\mathcal{B}}_{\lambda_2}, \ldots, \mathbf{s}^{\mathcal{B}}_{\lambda_{|\lambda^{\mathrm{set}}|}}\}$, that define how the target sparsity $s$ is distributed across blocks using a linear schedule in $[s - \lambda_k, s + \lambda_k]$. Formally, each $\mathbf{s}^{\mathcal{B}}_{\lambda_k}$ is a vector of length $|\mathcal{B}|$ where each element $\mathbf{s}^{\mathcal{B}}_{\lambda_k}(i)$ is the block-wise sparsity for the $i$-th block. Each vector $\mathbf{s}^{\mathcal{B}}_{\lambda_k}$ is computed using the following function:

$$\mathbf{s}^{\mathcal{B}}_{\lambda_k} = \mathtt{GetDist}(s, \lambda_k; \mathbf{v}) = \mathbf{1} - \left( 2\lambda_k \cdot \frac{\mathbf{v} - v_{\min}}{v_{\max} - v_{\min}} \; - \; \overline{g} \; + \; (1 - s) \right) \tag{4}$$

where $\mathbf{v} = [v_1, \ldots, v_{\mathcal{B}}]$ is a vector of block indices (with $v_{\min} = 1$ and $v_{\max} = |\mathcal{B}|$), and $\overline{g} = \mathrm{mean}\Big( 2\lambda_k \cdot \frac{\mathbf{v} - v_{\min}}{v_{\max} - v_{\min}} \Big)$. For the block-wise case, since we employed a linear schedule, we set $v_i$ to increase linearly with the block index $i$ (i.e., $v_i = i \; \forall i \in [1, |\mathcal{B}|]$), which yields $\mathbf{s}^{\mathcal{B}}_{\lambda_k}(i) = s - \lambda_k + 2\lambda_k \frac{i-1}{\mathcal{B}-1}$.

For each vector $\mathbf{s}^{\mathcal{B}}_{\lambda_k}$, we apply pruning at the block level using the base pruning method $\mathcal{P}$, and compute the alignment score from Eq. 3. We then select the best configuration as:

$$\mathbf{s}^{\mathcal{B}}_{\mathrm{best}} = \underset{\mathbf{s}_\lambda \in \{\mathbf{s}^{\mathcal{B}}_\lambda \,|\, \lambda \in \lambda^{\mathrm{set}}\}}{\arg\min} \; \mathtt{NeuronAL}\left(\mathcal{D}, \mathcal{S}, \mathcal{C}_\lambda\right) \tag{5}$$

where $\mathcal{S} = \mathcal{D} \odot \bigoplus_{i=1}^{|\mathcal{B}|} \bigoplus_{j=1}^{|\ell_i|} \mathtt{top}_{\mathbf{s}_\lambda}\big(\Psi^{\ell^j_i}, \mathbf{W}^{\ell^j_i}\big)$, with $\bigoplus$ indicating a mask concatenation operation. This sparsity configuration is then applied to the dense model to obtain $\mathcal{S}_{\mathcal{B}}$ that has the best alignment across blocks. This step captures variations in block importance without modifying individual neurons.

### 4.1.2 Row-Wise Sparsity Ratio

We then redistribute the sparsity ratios across the rows of each projection matrix. For each layer $\ell^j_i$ with weight matrix $\mathbf{W}^{\ell^j_i} \in \mathbb{R}^{r \times m}$, we generate a set of row-wise sparsity vectors $\mathbf{s}^r_{\lambda_k}$ using a linear schedule in $[s_{\mathcal{B}_i} - \lambda_k, s_{\mathcal{B}_i} + \lambda_k]$, with $\lambda_k \in \lambda^{\mathrm{set}}$. Different from the *block* step, the initial sparsity of layer $j$ is given from its parent block $i$ from the sparsified model $\mathcal{S}_{\mathcal{B}}$. Importantly, each $\mathbf{s}^r_{\lambda_k}$ defines a vector of row-wise sparsity values that is inversely proportional to the neuron alignment of each corresponding row, i.e., the rows that deviate more from the dense model are set as less sparse. Hence, the row-wise sparsity distribution is computed using the same function introduced in Eq. 4: $\mathbf{s}^r_{\lambda_k} = \mathtt{GetDist}(s_{\mathcal{B}_i}, \lambda_k; \mathbf{v}^{\mathrm{row}})$ where $\mathbf{v}^{\mathrm{row}}$ is the vector of neuron alignment values associated with the rows of $\mathbf{W}^{\ell^j_i}$. For each vector in set $\mathbf{s}^r_{\mathrm{set}} = \{\mathbf{s}^r_{\lambda_1}, \ldots, \mathbf{s}^r_{\lambda_{|\lambda^{\mathrm{set}}|}}\}$, we prune rows using the original saliency scores and compute the neuron alignment score from Eq. 3, applied over $\mathcal{S}_{\mathcal{B}}$. We then select the configuration that minimizes the neuron alignment metric:

$$\mathbf{s}^r_{\mathrm{best}} = \underset{\mathbf{s}^r_\lambda \in \{\mathbf{s}^r_\lambda \,|\, \lambda \in \lambda^{\mathrm{set}}\}}{\arg\min} \; \mathtt{NeuronAL}\left(\mathcal{D}, \mathcal{S}', \mathcal{C}_\lambda\right) \tag{6}$$

where $\mathcal{S}' = \mathcal{S}_{\mathcal{B}} \odot \bigoplus_{i=1}^{\mathcal{B}} \bigoplus_{j=1}^{\ell_i} \mathtt{top}_{\mathbf{s}^r_\lambda}\big(\Psi^{\ell^j_i}, \mathbf{W}^{\ell^j_i}\big)$ The full procedure, composed of block-wise and row-wise sparsity reallocation, is shown in Fig. 3, where $align = \mathrm{True}$ for $\mathtt{NeuroAl}(\cdot)$ returns, for each layer $\ell^j_i \in \mathbf{B}$ the row-wise alignment vector $\mathbf{v}^{\mathrm{row}}$ between $\mathcal{D}$ and $\mathcal{S}_{\mathcal{B}}$.

### 4.2 Non-Uniform Block-Wise Sparsity Distribution

Given the sparsity values for each block in $\mathbf{s}^{\mathcal{B}}_{\lambda_k}$ and a selected sparsity window $[s - \lambda, s + \lambda]$, the non-uniform block-wise sparsity schedule redistributes the sparsity across blocks in a monotonically *linear* way (i.e., the sparsity of block $i$ is always larger than the sparsity of layer $i-1, \forall i > 1$) via Eq. 5. We select this sparsity schedule for three main reasons: (1) as shown below such a straightforward sparsity schedule is already able to achieve similar results w.r.t. state-of-the art approaches, (2) to align with the latest discoveries in the literature of structured pruning where is consistently demonstrate how deeper blocks are redundant and can be removed with marginal performance degradation (Gromov et al., 2024; Men et al., 2024; Kim et al., 2024), and (3) to avoid having another sub-routine to select the best sparsity schedule, that, if linked with the block-wise and row-wise $\lambda$ selection, would have lead to have a combinatorial search space.

**Input:** $\mathcal{D}, \mathcal{P}, s, C_\lambda, \lambda^{\text{set}}$
```
// Block step
```
$\mathbf{v}^{\text{blk}} \leftarrow [1, \ldots, \mathcal{B}];$
$\left(\mathbf{s}_{set}^{\mathcal{B}}\right)^* \leftarrow \texttt{GetBestNeuronAL}(\mathcal{D}, s, \lambda^{\text{set}}, C_\lambda, \mathcal{P}, \mathbf{v}^{\text{blk}});$
$\mathcal{S}_{\mathcal{B}} \leftarrow \mathcal{D} \odot \bigoplus_{i=1}^{\mathcal{B}} \texttt{top}_{(\mathbf{s}_{set}^{\mathcal{B}})^*}\left(\Psi^{\ell_i}, \mathbf{W}^{\ell_i}\right);$
```
// Row step
```
$\mathbf{v}^{\text{row}} \leftarrow \texttt{NeuronAL}(\mathcal{D}, \mathcal{S}_{\mathcal{B}}, C_\lambda;\ align=\text{True});$
$\left(\mathbf{s}_{set}^{r}\right)^* \leftarrow \texttt{GetBestNeuronAL}(\mathcal{D}, s, \lambda^{\text{set}}, C_\lambda, \mathcal{P}, \mathbf{v}^{\text{row}});$
$\mathcal{S}_{\text{final}} \leftarrow \mathcal{D} \odot \bigoplus_{i=1}^{\mathcal{B}} \bigoplus_{j=1}^{l_i} \texttt{top}_{(\mathbf{s}_{set}^{r})^*}\left(\Psi^{\ell_i^j}, \mathbf{W}^{\ell_i^j}\right)$

**Function**
$\texttt{GetBestNeuronAL}(\mathcal{D}, s, \lambda^{set}, C_\lambda, \mathcal{P}, \mathbf{v})\texttt{:}$
  $\mathbf{s}^* \leftarrow \emptyset;$
  $neur_{al}^* \leftarrow \infty;$
  $\Psi = \mathcal{P}(\mathcal{D});$
  **foreach** $\lambda \in \lambda^{set}$ **do**
    $\mathbf{s}_\lambda \leftarrow \texttt{GetDist}(s, \lambda, \mathbf{v})$ ; // Eq. 4
    $\mathcal{S} \leftarrow \left(\mathcal{D} \odot \bigoplus_{i=1}^{\mathcal{B}} \bigoplus_{j=1}^{l_i} \texttt{top}_{\mathbf{s}_\lambda}\left(\Psi^{\ell_i^j}, \mathbf{W}^{\ell_i^j}\right)\right)$
    $neur_{al} \leftarrow \texttt{NeuronAL}(\mathcal{D}, \mathcal{S}, C_\lambda)$ ; // Eq. 3
    **if** $neur_{al} < neur_{al}^*$ **then**
      $\mathbf{s}^* \leftarrow \mathbf{s}_\lambda;$
      $neur_{al}^* \leftarrow neur_{al};$
    **end**
  **end**
  **return** $\mathbf{s}^*;$

Figure 3: Left: Overall NEURONAL top-up pruning procedure. Right: GETBESTNEURONAL sub-routine used in both block- and row-selection stages.

We motivated the choice of a linear schedule by testing three straightforward non-uniform sparsity schedules (namely *linear*, *exponential*, and *logarithmic*), which do not require any block scoring for sparsity allocation. Table 2 displays the improvement, w.r.t. uniform distribution (averaged across three different Language Modeling datasets, namely Wiki-Text2, C4, PTB), achieved by the three sparsity schedules using Wanda as pruning algorithm with $\lambda = 0.08$. The results highlight how non-uniform sparsity schedules, without any block-based scoring, lead to a performance

Table 2: Performance improvement w.r.t. uniform distribution averaged across three different datasets (WikiText2, C4, and PTB) using Wanda as pruning algorithm.

| Sparsity | Model | Schedule | | | |
|---|---|---|---|---|---|
| | | OWL | Exp | Log | Linear |
| 60% | Phi-2.7B | +3.4% | +2.4% | **+7.8%** | +7.7% |
| | LLama-1 7B | +16.5% | +3.4% | +15.7% | **+18.1%** |
| 70% | Phi-2.7B | +45.8% | +47.7% | +44.9% | **+52.5%** |
| | LLama-1 7B | **+66.8%** | +28.2% | +53.9% | +63.5% |
| 80% | Phi-2.7B | +87.8% | **+89.3%** | +55.7% | +82.8% |
| | LLama-1 7B | **+81.5%** | +63.6% | -4.4% | +68.1% |
| Mean | | **+50.3%** | +39.1% | +28.9% | +48.8% |

improvement close to OWL's. Overall, the *linear* schedule turns out to be the most reliable one since it does not show oscillations in performance across the different sparsity ratios (while this happens for the logarithmic and exponential schedules).

## 5 Experiments

We apply our proposed NEURONAL to different state-of-the-art pruning algorithms tailored for LLMs. Specifically, we test how it compares in terms of performance over Language Modeling datasets and Zero-Shot tasks w.r.t. the most recent top-up algorithms for pruning. We also perform scalability and sensitivity analyses to show the effectiveness of our NEURONAL.

### 5.1 Experimental Setup

**Language Modeling Datasets** To measure the models' perplexity on Language Modeling datasets, we use the following three datasets: (1) *WikiText2* (Merity et al., 2017), (2) Colossal Clean Common Crawl (*C4*) (Raffel et al., 2020), and (3) Penn Treebank (*PTB*).

**Zero-Shot Tasks** To assess more thoroughly how the different pruning algorithms affect the models' capabilities, we employ the following 7 datasets: (1) Recognizing Textual Entailment (*RTE*) (Dagan et al., 2006; Bar Haim et al., 2006; Giampiccolo et al., 2007; Bentivogli et al., 2009) , (2) *WinoGrande* (Sakaguchi et al., 2021), (3) *BoolQ* (Clark et al., 2019), (4) *HellaSwag* (Zellers et al., 2019), (5) *ARC-e* (Clark et al., 2018), (6) *ARC-c* (Clark et al., 2018), (7) *OBQA* (Mihaylov et al., 2018)

**Models and Sparsity** Since one of the distinctive features of NEURONAL is its adaptability w.r.t. sparsity and models, we test four different LLM families, namely LLama 7B (both v1 and v2) (Touvron et al.,

Table 3: Perplexity on the three Language Modeling datasets computed over five different LLMs for four different top-up algorithms (Uniform, DSnoT, OWL, and NEURONAL) on three pruning algorithms (Magnitude, MULTIFLOW, and Wanda) at 70% sparsity.

| Algorithm | Top-Up | Phi-2.7B | | | LLama-1 7B | | | LLama-2 7B | | | Mistral-7B | | | OPT-6.7B | | |
|---|---|---|---|---|---|---|---|---|---|---|---|---|---|---|---|---|
| | | WikiText2 | C4 | PTB | WikiText2 | C4 | PTB | WikiText2 | C4 | PTB | WikiText2 | C4 | PTB | WikiText2 | C4 | PTB |
| Dense | | 9.7 | 14.1 | 18.2 | 5.7 | 7.3 | 10.1 | 5.5 | 7.3 | 32.9 | 5.3 | 8.4 | 36.6 | 10.9 | 12.7 | 15.8 |
| Magnitude | Uniform | 764.6 | 384.4 | 983.9 | 2.53e4 | 2.25e4 | 3.26e4 | 1.42e5 | 1.02e4 | 2.02e6 | 221.9 | 232.9 | 748.7 | 1.00e4 | 5.39e3 | 6.54e3 |
| | DSnoT | 539.0 | 258.0 | 656.2 | 1.02e7 | 2.77e6 | 4.99e7 | 1.31e8 | 2.90e7 | 2.25e8 | 192.7 | 189.9 | 566.2 | **6.16e3** | **3.93e3** | **4.36e3** |
| | OWL | 419.6 | 242.7 | 358.5 | 1.20e4 | 6.58e3 | 5.39e4 | 3.39e5 | 1.24e4 | 3.28e6 | 111.7 | 124.2 | 545.5 | 1.57e4 | 8.48e3 | 9.67e3 |
| | AlphaPruning | 2.52e4 | 1.60e4 | 2.34e4 | 424.9 | 391.5 | 5.08e4 | 3.37e3 | 3.60e3 | 1.73e5 | 91.3 | 106.5 | 717.1 | 1.22e4 | 7.22e3 | 7.51e3 |
| | NEURONAL | **281.7** | **180.9** | **321.1** | **231.8** | **219.9** | **4.46e3** | **155.8** | **264.8** | **2.61e3** | **46.5** | **43.1** | **612.8** | 2.11e4 | 1.07e4 | 1.09e4 |
| MULTIFLOW | Uniform | 388.4 | 298.8 | 610.8 | 80.9 | 71.9 | 172.4 | 60.0 | 58.8 | 1.26e3 | 9.37e2 | 6.56e2 | 2.06e3 | 9.44e2 | 1.25e3 | 843.1 |
| | DSnoT | 325.5 | 261.9 | 328.8 | 67.6 | 65.0 | 114.7 | 66.6 | 75.8 | 6.89e2 | **57.4** | **63.3** | **2.65e2** | 241.8 | 153.3 | 263.9 |
| | OWL | 197.9 | 141.3 | 293.9 | 25.1 | 25.8 | 78.9 | 29.2 | 31.0 | 5.47e2 | 329.0 | 7.64e2 | 1.72e3 | 240.9 | 495.6 | 337.8 |
| | AlphaPruning | 1.22e5 | 8.99e4 | 9.52e4 | 32.2 | 35.2 | 103.8 | 31.3 | 34.0 | 287.3 | 230.8 | 292.8 | 1.72e3 | **133.8** | **63.7** | 153.9 |
| | NEURONAL | **105.4** | **87.1** | **179.5** | **20.7** | **21.2** | **46.2** | **22.1** | **23.9** | 265.5 | _202.5_ | 334.7 | _1.41e3_ | _209.7_ | _83.7_ | _202.1_ |
| Wanda | Uniform | 227.6 | 182.7 | 346.2 | 85.1 | 86.2 | 157.0 | 78.0 | 81.0 | 599.3 | 60.7 | 73.6 | 298.3 | 157.5 | 260.1 | 209.2 |
| | DSnoT | 221.9 | 172.6 | 257.6 | 72.9 | 76.0 | 121.0 | 76.1 | 85.7 | 491.8 | 81.3 | 79.9 | 304.8 | 191.4 | 173.3 | 182.6 |
| | OWL | 132.7 | 116.2 | 183.7 | 24.6 | 27.3 | 61.2 | 30.5 | 36.6 | 333.7 | 41.0 | 51.8 | 253.5 | **54.4** | **69.7** | **100.7** |
| | AlphaPruning | 4.22e4 | 3.05e4 | 2.23e4 | 26.9 | 31.1 | 77.4 | 32.0 | 37.7 | 273.8 | 39.4 | 49.8 | 286.8 | 93.8 | 53.7 | 120.9 |
| | NEURONAL | **88.3** | **77.7** | **129.5** | **21.5** | **23.2** | **44.2** | **24.0** | **27.4** | **207.0** | **28.8** | **33.7** | **232.0** | 172.6 | 84.0 | 182.7 |

2023a;b), Phi-2, Mistral-7B (Jiang et al., 2023), and OPT-6.7B (Zhang et al., 2022). To scale up the model size, we also test LLama 13B (both v1 and v2) and LLama 70B (v2). In the paper, we mainly present results at 60%, 70%, and 80% sparsity, for fair comparisons with (Yin et al., 2024; Lu et al., 2024). To assess the generalization to different sparsity ratios, we present the results on broader sparsity ratios (from 10% to 80%) in Appendix D.7.

**Baselines** As pruning algorithms, we test Magnitude, MULTIFLOW (Farina et al., 2024), and Wanda (Sun et al., 2023). All are tested with four different top-up algorithms (besides ours): (1) **Uniform** distribution, (2) **DsnoT** (Zhang et al., 2024) (dynamic training-free uniform distribution with mask update), (3) **OWL** (Yin et al., 2024) (block-wise training-free non-uniform distribution based on outliers scores) and (4) **AlphaPruning** (Lu et al., 2024) (block-wise training-free non-uniform distribution based on ESD)[3]. Further details on the setup are in Appendix C.

Table 4: Average accuracy on the seven Zero-Shot tasks using Wanda as pruning algorithm.

| Sparsity | Top-Up | Model | | | | |
|---|---|---|---|---|---|---|
| | | Phi-2.7B | LLama-1 7B | LLama-2 7B | Mistral-7B | OPT-6.7B |
| Dense | | 64.35 | 59.97 | 59.71 | 64.21 | 51.52 |
| 60% | Uniform | **52.36** | 50.39 | 49.97 | 51.03 | **46.24** |
| | DSnoT | 49.36 | 49.12 | 48.83 | 50.51 | 45.75 |
| | OWL | 51.48 | 50.93 | 51.68 | **52.49** | 46.05 |
| | AlphaPruning | 43.14 | 51.08 | 51.21 | 49.87 | 44.50 |
| | NEURONAL | _52.04_ | **51.41** | **51.99** | _52.09_ | _46.10_ |
| 70% | Uniform | 39.89 | 36.90 | 34.37 | 36.85 | 36.31 |
| | DSnoT | 38.76 | 36.26 | 34.09 | 36.53 | 36.35 |
| | OWL | 41.20 | 43.31 | 40.57 | 38.77 | 38.77 |
| | AlphaPruning | 35.02 | 44.08 | 42.03 | 39.05 | **39.53** |
| | NEURONAL | **41.87** | **44.57** | **43.10** | **41.56** | _38.86_ |
| 80% | Uniform | 36.21 | 31.66 | 31.81 | 32.48 | 33.34 |
| | DSnoT | 32.73 | 31.78 | 32.27 | 32.14 | **35.12** |
| | OWL | 36.26 | 31.43 | 32.48 | 32.02 | 32.10 |
| | AlphaPruning | 30.74 | 35.34 | 32.09 | 32.29 | 32.16 |
| | NEURONAL | **37.36** | **36.31** | **32.74** | **33.08** | 33.05 |

## 5.2 Experimental Evaluation

In this section, we show the numerical results of our proposed NEURONAL w.r.t. the baselines for Language Modeling and Zero-Shot tasks.

### 5.2.1 Language Modeling and Zero-shot Tasks

Concerning the Language Modeling datasets, the numerical results in terms of perplexity computed over the three Language Modeling datasets at 70% sparsity are shown in Table 3. It can be seen how NEURONAL is

---

[3]BESA and DSA are not included in these experiments due their large runtime. Testing them on all combinations of sparsities and pruning algorithms is unfeasible with our GPU resources.

able in almost all cases to outperform all the other baselines by a large margin. In no case does NEURONAL perform worse w.r.t. the uniform distribution. The only model for which NEURONAL is not the best top-up algorithm for all pruning algorithms is OPT. In all other cases, NEURONAL outperforms OWL and AlphaPruning for all models and pruning algorithms. The results at 60% and 80% sparsity shown in Tables 17-18 in the Appendix confirm this trend.

As for the Zero-Shot tasks, the numerical results are shown in Tables 4-5[4]. We display only the mean across the seven Zero-Shot tasks, while the results for each task are available in Tables 19-21 in the Appendix. Again, NEURONAL turns out to outperform in the majority of cases all the baselines. In 20 cases out of 30 (w.r.t. the mean accuracy across all tasks), NEURONAL is the one that reaches the best performance, and in 5 cases, the second best.

Table 5: Average accuracy on the seven Zero-Shot tasks using MULTIFLOW as pruning algorithm.

| Sparsity | Top-Up | Model | | | | |
|---|---|---|---|---|---|---|
| | | Phi-2.7B | LLama-1 7B | LLama-2 7B | Mistral-7B | OPT-6.7B |
| Dense | | 64.35 | 59.97 | 59.71 | 64.21 | 51.52 |
| 60% | Uniform | 53.34 | 49.60 | 48.76 | 32.51 | 45.76 |
| | DSnoT | 48.90 | 49.57 | 49.62 | **51.93** | 45.75 |
| | OWL | 53.04 | 49.70 | 49.12 | 34.70 | **46.15** |
| | AlphaPruning | 40.25 | 49.13 | 48.95 | 34.97 | 45.01 |
| | NEURONAL | **53.77** | **50.57** | **51.08** | 34.07 | 45.51 |
| 70% | Uniform | 38.40 | 38.51 | 36.98 | 32.76 | 33.91 |
| | DSnoT | 38.76 | 37.51 | 36.25 | **37.83** | 36.33 |
| | OWL | 42.46 | 43.23 | 40.60 | 32.63 | 35.69 |
| | AlphaPruning | 31.81 | 44.07 | 41.50 | 31.95 | 37.83 |
| | NEURONAL | **43.18** | **45.93** | **42.87** | 32.63 | **39.34** |
| 80% | Uniform | 34.32 | 32.10 | 32.63 | 31.98 | 32.34 |
| | DSnoT | 32.20 | 31.45 | 32.23 | 32.18 | 34.84 |
| | OWL | **35.99** | 33.25 | 32.19 | 32.04 | 32.67 |
| | AlphaPruning | 30.72 | 36.42 | 32.30 | **32.77** | 32.65 |
| | NEURONAL | 34.86 | **37.61** | **32.99** | 32.32 | **36.63** |

## 5.3 Aggregate Comparison of *Top-up* Strategies

While Tables 3-5 already show that NEURONAL consistently outperforms the baselines, one of the main strengths of our proposed method lies in its ability to adapt to any sparsity, model, pruning algorithm, and tested dataset. For showing that, we design two metrics that aggregate, for each tested *top-up* algorithm, the results at sparsity ratios $s \in \{60, 70, 80\}\%$ by calculating the geometric mean of the results of different models ($LLM$), pruning methods ($P$) and datasets ($D$) at each sparsity. The two metrics are: (1) the *accuracy retention*, defined as:

Table 6: Summary of *accuracy retention* (% w.r.t. dense) and *perplexity degradation factor* ($\times$ dense) at 60%, 70%, and 80% sparsity for up to 7B models.

| Top-Up | Acc. retention (%) ↑ | | | PPL factor (×)↓ | | |
|---|---|---|---|---|---|---|
| | 60% | 70% | 80% | 60% | 70% | 80% |
| Uniform | 79.6 | 61.0 | 55.0 | 9.0 | 62.2 | 786.7 |
| DSnoT | 81.8 | 61.6 | 54.7 | 7.0 | 96.9 | 746.3 |
| OWL | 80.7 | 66.2 | 55.2 | 6.1 | 35.5 | 392.4 |
| AlphaPruning | 76.2 | 64.3 | 54.7 | 8.3 | 66.0 | 505.7 |
| **NeuronAL** | **81.1** | **69.0** | **58.0** | **4.1** | **14.6** | **201.8** |

$$\text{AccRet}(s) = \Big( \prod_{m \in \mathcal{LLM}} \prod_{p \in \mathcal{P}} \frac{\text{Acc}_{s,p}(m)}{\text{Acc}_{\text{dense}}(m)} \Big)^{1/(|\mathcal{LLM}| |\mathcal{P}|)} \tag{7}$$

and (2) the *perplexity factor*, defined as:

$$\text{PPLFac}(s) = \Big( \prod_{m \in \mathcal{LLM}} \prod_{d \in \mathcal{D}} \prod_{p \in \mathcal{P}} \frac{\text{PPL}_{s,p}(m,d)}{\text{PPL}_{\text{dense}}(m,d)} \Big)^{1/(|\mathcal{LLM}| |\mathcal{D}| |\mathcal{P}|)} . \tag{8}$$

---

[4]Magnitude is omitted from these tables, due to space constraints. Results are available in Tables 22-24 in the Appendix.

The data used to compute the values shown in Table 6 are as follows. For the *accuracy retention*, we used the average performance across zero-shot tasks and took the data from Tables 4-5; for the *perplexity factor* we took the perplexity results from Table 3 and Tables 17-18 reported in the Appendix.

The results clearly show that, when aggregating all the performance across pruning methods, datasets, and language models (in the order of 7B scale), NEURONAL always achieves better performance w.r.t. the *top-up* baselines. These results, apart from providing additional evidence about the robustness of our approach, also highlight its adaptability as one of the main strengths.

### 5.3.1 Scalability Study

To assess if the NEURONAL performance scales to bigger models, we apply it to LLama 13B (v1 and v2) and LLama 70B (v2) on both the Language Modeling datasets and Zero-Shot tasks. Also in this case, we provide, see Table 7, the aggregated results using the *perplexity factor* and *accuracy retention* defined in Section 5.3. These aggregated results clearly show that, on average, NEURONAL is in line with AlphaPruning in terms of perplexity, but outperforms it in the Zero-Shot tasks, even when scaling up the LLM size, while requiring 20× less time to obtain the non-uniform sparsity

Table 7: Summary of top-up *accuracy retention* (% w.r.t. dense) and *perplexity degradation factor* (× dense) at 70% and 80% sparsity for 13B and 70B models.

| | Acc. retention (%) ↑ | | PPL factor ↓ | |
|---|---|---|---|---|
| **Top-Up** | 70% | 80% | 70% | 80% |
| Uniform | 67.9 | 52.2 | 13.1 | 219.4 |
| DSnoT | 67.5 | 51.5 | 6.3 | 229.6 |
| OWL | 77.8 | 53.7 | 6.0 | 82.8 |
| AlphaPruning | 78.8 | 60.4 | 3.8 | **35.9** |
| **NeuronAL** | **80.9** | **62.4** | **3.6** | 47.0 |

schedule. Looking at the detailed results in Table 8-9, it further becomes clearer that NEURONAL and AlphaPruning both provide advantages in this experimental setting. When evaluating over Language Modeling datasets, see Table 8, our proposed approach provides better performance over the 13B models, while AlphaPruning over the 70B case. On the other hand, when evaluating over Zero-Shot tasks, the mean accuracy across the seven tasks shows that NEURONAL outperforms AlphaPruning in 10 out of 12 cases, see shown in Table 9 (for the individual tasks' results see Appendix D.5). However, it is worth mentioning that, in contrast with AlphaPruning, which in the 70B case requires hours (∼12 hours on an A100 80GB) for computing the non-uniform sparsity schedule, NEURONAL requires ∼35 minutes to obtain it, as shown in Table 10.

Table 8: Perplexity of LLama models (v1 of 13B and v2 of both 13B and 70B) on the three Language Modeling datasets at 70% (top) and 80% sparsity (bottom).

| Algorithm | Top-Up | LLama-1 13B | | | LLama-2 13B | | | LLama-2 70B | | |
|---|---|---|---|---|---|---|---|---|---|---|
| | | WikiText-2 | C4 | PTB | WikiText-2 | C4 | PTB | WikiText-2 | C4 | PTB |
| Dense | | 5.09 | 6.80 | 28.11 | 4.88 | 6.73 | 48.82 | 3.32 | 5.71 | 20.76 |
| MULTIFLOW | Uniform | 49.4 | 45.3 | 277.8 | 144.3 | 112.4 | 623.2 | 11.59 | 15.09 | 216.34 |
| | DSnoT | 46.2 | 48.9 | 240.4 | 45.8 | 54.2 | 611.5 | 9.49 | 13.22 | 86.17 |
| | OWL | 16.6 | 17.7 | 132.2 | 54.0 | 56.2 | 426.6 | 9.32 | 12.13 | 85.16 |
| | AlphaPruning | **13.8** | 16.1 | 132.6 | 26.9 | 32.6 | 337.9 | **7.98** | **10.91** | **47.20** |
| | NEURONAL | **13.8** | **15.6** | **101.7** | **20.1** | **23.1** | **318.9** | 9.03 | 11.75 | 120.84 |
| Wanda | Uniform | 54.4 | 55.3 | 309.2 | 45.7 | 56.2 | 571.0 | 10.59 | 14.17 | 88.01 |
| | DSnoT | 47.8 | 54.2 | 248.6 | 46.6 | 57.7 | 555.5 | 10.14 | 13.97 | 74.87 |
| | OWL | 16.3 | 18.9 | 147.6 | 18.0 | 21.8 | 315.1 | 9.01 | 11.92 | 54.96 |
| | AlphaPruning | 14.6 | 17.3 | 126.7 | **15.2** | **18.8** | 271.1 | **7.82** | **10.78** | **39.21** |
| | NEURONAL | **14.3** | **16.6** | **97.9** | 16.5 | 19.3 | **237.6** | 8.46 | 11.18 | 46.66 |
| MULTIFLOW | Uniform | 3.71e3 | 1.70e3 | 3.59e3 | 4.48e3 | 2.41e3 | 5.21e3 | 266.20 | 175.07 | 956.49 |
| | DSnoT | 5.37e3 | 2.86e3 | 6.29e3 | 1.94e3 | 1.67e3 | 5.28e3 | 189.76 | 132.22 | 698.30 |
| | OWL | 813.8 | 375.7 | 2.14e3 | 1.80e3 | 1.01e3 | 4.39e3 | 84.52 | 74.02 | 727.40 |
| | AlphaPruning | 210.7 | 147.8 | 1.19e3 | **458.1** | **279.7** | **1.93e3** | **33.69** | **37.23** | **241.23** |
| | NEURONAL | **126.8** | **123.7** | **901.6** | 1.60e3 | 864.1 | 2.96e3 | 52.73 | 44.07 | 747.53 |
| Wanda | Uniform | 3.48e3 | 1.96e3 | 3.57e3 | 1.12e3 | 870.5 | 5.55e3 | 151.80 | 122.17 | 606.57 |
| | DSnoT | 4.37e4 | 2.44e4 | 3.22e4 | 4.44e3 | 3.96e3 | 4.09e3 | 193.28 | 137.61 | 620.97 |
| | OWL | 761.6 | 368.1 | 1.93e3 | 248.0 | 204.2 | 2.03e3 | 56.07 | 59.57 | 368.97 |
| | AlphaPruning | 209.6 | 148.6 | **973.3** | 165.1 | 158.2 | 1.53e3 | **31.10** | **36.19** | **162.65** |
| | NEURONAL | **156.4** | **132.4** | 1.47e3 | 185.7 | **143.1** | **1.25e3** | 47.72 | 40.28 | 212.02 |

Table 9: Average accuracy on the seven Zero-Shot tasks using both Wanda and MULTIFLOW as pruning algorithm on larger-scale LLama models (13B, both v1 and v2, and 70B v2) at 70% (top) and 80% (bottom) sparsity.

| Sparsity | Top-Up | LLama-1 13B | | LLama-2 13B | | LLama-2 70B | |
|---|---|---|---|---|---|---|---|
| | | Wanda | MULTIFLOW | Wanda | MULTIFLOW | Wanda | MULTIFLOW |
| Dense | | 62.64 | | 63.03 | | 67.10 | |
| 70% | Uniform | 38.95 | 40.43 | 37.54 | 32.73 | 56.73 | 55.02 |
| | DSnoT | 38.88 | 39.93 | 37.34 | 36.96 | 56.09 | 57.36 |
| | OWL | 46.52 | 47.31 | 46.33 | 38.05 | 57.91 | 56.60 |
| | AlphaPruning | 47.41 | 48.22 | 47.46 | 44.95 | 57.62 | 56.83 |
| | NEURONAL | **49.34** | **50.79** | **48.28** | **47.82** | **58.96** | **56.95** |
| 80% | Uniform | 32.76 | 32.78 | 32.82 | 32.4 | 34.98 | 35.18 |
| | DSnoT | 32.41 | 32.43 | 32.14 | 32.59 | 34.65 | 34.82 |
| | OWL | 32.25 | 32.26 | 32.49 | 32.36 | 39.21 | 38.18 |
| | AlphaPruning | **38.07** | 38.56 | **36.59** | 32.28 | 42.00 | 41.63 |
| | NEURONAL | 37.98 | **40.72** | 36.19 | **32.51** | **46.79** | **45.32** |

## 5.4 Efficiency Analysis

In this section, we analyze the efficiency of NEURONAL in terms of pruning runtime and inference speed-up.

### 5.4.1 Runtime vs. Perplexity

NEURONAL provides a good trade-off between performance and runtime. In Table 11, we show for all baselines (here we also include BESA and DSA), the runtime in seconds required to obtain the non-uniform sparsity distribution for the given model (in this case, LLama-1 7B) as well as the performance computed as the perplexity over WikiText2. The results confirm how NEURONAL can achieve, in 3 out of 4 cases, the best results in terms of perplexity while maintaining a low computational budget. In terms of runtime, the only comparable methods are DSnoT and OWL, compared to which, however, NEURONAL achieves better performance. On the other hand, DSA is the closest in terms of perplexity to NEURONAL, while requiring four orders of magnitude more time to obtain the best sparsity distribution. Overall, the performance-runtime trade-off of NEURONAL improves when increasing the sparsity ratio.

Moreover, in Table 10 we report the runtime of the baselines used as comparison in the paper over the 13B and 70B LLama models. As expected, the results of the runtime scale up with the model's size. Overall, DsNoT and OWL display the lowest runtime, while NEURONAL tends to be ∼3.5×-4× slower, even when scaling the model size. On the other hand, NEURONAL is drastically faster than AlphaPruning, from 6× in the 7B case, up to ∼20× in the 70B case. This runtime advantage indicates that NEURONAL not only provides better performance, which is especially evident on larger models (see Tables 8-9), but also achieves a better performance-runtime trade-off.

Table 10: Runtime (seconds) for obtaining the non-uniform sparsity allocation among different *top-up* algorithms over LLama 13B and 70B at 70% sparsity using Wanda.

| Metric | Top-up pruning algorithms | | | |
|---|---|---|---|---|
| | DsNoT | OWL | AlphaPruning | NeuronAL |
| LLama-1 13B | 129.61s | 140.55s | 5443.40s | 525.81s |
| LLama-2 13B | 129.36s | 140.46s | 5379.42s | 524.41s |
| LLama-2 70B | 417.92s | 488.76s | 41890.97s | 2110.20s |

Table 11: Runtime (seconds) vs. perplexity trade-off comparison among different *top-up* algorithms over LLama-1 7B pruned at different sparsity ratios using Wanda.

| Metric | Top-up pruning algorithms | | | | | | |
|---|---|---|---|---|---|---|---|
| | Uniform | DsNoT | OWL | BESA | DSA | AlphaPruning | NeuronAL |
| Runtime | - | 4.5s | 73.3s | $\sim 1.8 \times 10^4$ s | $\sim 4.3 \times 10^4$ s | 1479.4s | 237.1s |
| Perplexity @ 65% | 20.9 | 19.1 | 13.1 | 18.5 | **12.6** | 14.0 | 12.8 |
| Perplexity @ 70% | 85.1 | 72.9 | 24.6 | 42.6 | 22.6 | 26.9 | **20.7** |
| Perplexity @ 75% | 927.4 | 646.7 | 152.5 | 257.9 | 103.3 | 110.2 | **61.2** |
| Perplexity @ 80% | 5.22e3 | 3.71e3 | 986.5 | 2.21e3 | 736.81 | 768.4 | **302.8** |

### 5.4.2 Inference Speed-up

Here, we evaluate the speed-up acceleration of both dense and sparse models obtained with NEURONAL, using the same inference pipeline based on DeepSparse (NeuralMagic, 2021) ON-NXRuntime backends. The evaluation consists of the end-to-end token generation and has been done over an Intel i9-10980XE CPU using 18 cores. Table 12

Table 12: End-to-end inference speed-up (throughput gain) for Phi-2 and LLama-2 7B (v1) at different sparsity ratios. Throughput is measured in tokens/sec.

| Model | Metric | Dense | 20% | 40% | 60% | 80% |
|---|---|---|---|---|---|---|
| **Phi-2** | Throughput ↑ (tokens/s) | 0.1306 | 0.1307 | 0.1325 | 0.1488 | 0.1770 |
| | Speed-up ↑ | 1.00× | 1.00× | 1.01× | 1.14× | 1.35× |
| **LLama-2 7B** | Throughput ↑ (tokens/s) | 0.1498 | 0.1506 | 0.1554 | 0.1714 | 0.2309 |
| | Speed-up ↑ | 1.00× | 1.01× | 1.04× | 1.14× | 1.54× |

shows the throughput, measured in terms of tokens generated per second over Phi-2, using a sequence length of 2024, and LLama-2 7B, using a sequence length of 1024, with four different sparsity ratios, along with the speed-up w.r.t. the generation time required by the dense model.

## 5.5 Ablation Studies

In this section, we provide a set of ablation studies that further show the robustness of our proposed method.

### 5.5.1 NeuronAL $\lambda$ Selection

In this section, we report an analysis of the ability of NEURONAL to pick the best $\lambda$ parameters (i.e., the parameters for which the performance is the best one, hence the lowest value if computed over perplexity). To do this, we evaluate NEURONAL for all the $\lambda$ parameters (in the *block*-only setting, to simplify the visualization of results) over the three Language Modeling datasets. Fig. 4 reports the perplexity at 70% sparsity across different values of $\lambda$ (black dots connected by solid lines), while the dots highlighted in orange indicate the perplexity achieved with the $\lambda$ value selected by NEURONAL. These results highlight how NEURONAL, in the majority of the cases, can pick the best value of $\lambda$ both with data knowledge, as in the C4 dataset (from which the calibration data is sampled), as well as on unseen datasets such as WikiText2 and PTB. Figs 6-7 in the Appendix show the results at 60%-80% sparsity.

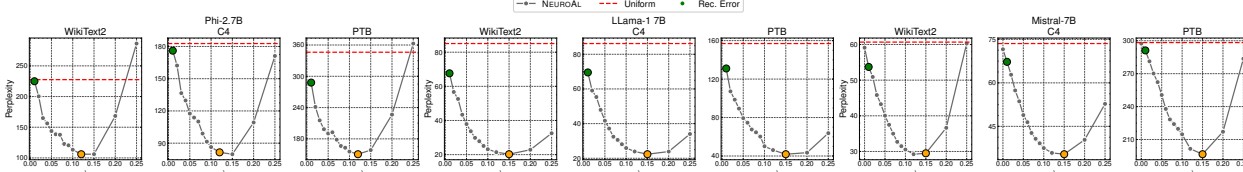

Figure 4: Perplexity over different values of $\lambda$ at 70% sparsity. The orange dot indicates the value selected by NEURONAL using neuron alignment. The green dot indicates the value selected by NEURONAL using the reconstruction error rather than the neuron alignment (see Section 5.5.2).

### 5.5.2 NeuronAL vs. Reconstruction Error and Distance Metrics

Our proposed approach is based on the *activation* alignment, defined in Eq.s 2-3. In Appendix A, we provide a detailed comparison of the formal difference between the definition of *neuron alignment* and that of Reconstruction Error (Rec-Error) (Frantar & Alistarh, 2023). In this section, we demonstrate, empirically, how such a difference translates into an improved performance of *neuron alignment* w.r.t. Rec-Error. To do so, we conduct an ablation study of NEURONAL where the best $\lambda$ for the block and row cases is selected based on the values that minimize the Rec-Error rather than the neuron alignment.

Table 13 shows the results, in terms of perplexity over WikiText2, of NEURONAL for the standard case with neuron alignment and the setting based on Rec-Error. As clearly visible, *neuron alignment* always leads to better performance. In addition, we highlight in Fig. 4 the $\lambda$ value (for the block-only case) selected using Rec-Error. As displayed in the figure, while NEURONAL picks the $\lambda$ that minimizes the *neuron alignment*, which in this case correlates with the lower (hence better) perplexity performance, Rec-Error does not provide the same ability.

Moreover, we include in the same table an additional ablation study about the *neuron alignment* formulation (Eq. 2). Specifically, we tested two different variations over Eq. 2 where the L2-norm is replaced by Cosine-Similarity (Cosine) and Kullback–Leibler (KL) divergence, respectively. In both cases, the procedure for the $\lambda$ selection remains the same as for NEURONAL. The results in Table 13 show that the original NEURONAL formulation with the L2-norm provides, in the majority of the cases (10 out of 12), the best results in terms of perplexity over WikiText2. This further ablation confirms the effectiveness of the proposed formulation.

### 5.5.3 Sensitivity to Calibration Data

Since NEURONAL works uses a calibration set, we test its sensitivity on the calibration settings in terms of seed, data source, and size. The results in Table 14 show the mean and standard deviation of perplexity at 70% sparsity over the three Language Modeling datasets for different seeds and when using different datasets to extract the calibration data. Furthermore, the results in Fig. 5

Table 13: Perplexity of NEURONAL vs. Rec-Error and different distance metrics (Cosine Similarity and KL-Divergence) across three sparsity ratios and four LLMs.

| Model | Top-Up | 60% | 70% | 80% |
|---|---|---|---|---|
| **LLama-1 7B** | Rec-Error | 10.78 | 79.40 | 3.29e3 |
| | NEURONAL w. Cosine | 9.66 | 25.01 | 536.53 |
| | NEURONAL w. KL | 10.42 | 70.99 | 6.18e3 |
| | NEURONAL | **9.59** | **21.53** | **302.82** |
| **LLama-2 7B** | Rec-Error | 10.96 | 67.25 | 2.32e3 |
| | NEURONAL w. Cosine | 9.56 | 27.93 | **448.76** |
| | NEURONAL w. KL | 10.36 | 66.93 | 1.54e3 |
| | NEURONAL | **9.38** | **24.05** | 557.17 |
| **Phi-2** | Rec-Error | **24.21** | 202.73 | 1.042e4 |
| | NEURONAL w. Cosine | 24.57 | 106.18 | 3.03e3 |
| | NEURONAL w. KL | 24.99 | 237.23 | 1.32e4 |
| | NEURONAL | 25.26 | **88.32** | **2.49e3** |
| **Mistral-7B** | Rec-Error | 10.95 | 56.07 | 290.73 |
| | NEURONAL w. Cosine | 10.78 | 33.52 | **214.54** |
| | NEURONAL w. KL | 10.97 | 55.01 | 341.45 |
| | NEURONAL | **9.94** | **28.78** | 249.47 |

report the perplexity at 70% sparsity over the three Language Modeling datasets when changing the number of calibration samples (using C4 as calibration source and 0 as seed).

Table 14: Perplexity (avg. $\pm$ std. dev.) achieved when using NEURONAL with different calibration data seeds (top), and when using different datasets as the source for the calibration data (bottom) on the three Language Modeling datasets at 70% sparsity.

| Calibration source(s), seed(s) | Dataset | Model | | | | |
|---|---|---|---|---|---|---|
| | | **Phi-2.7B** | **LLama-1 7B** | **LLama-2 7B** | **Mistral-7B** | **OPT-6.7B** |
| Calibration: {C4} Seeds: {0, 16, 46} | WikiText2 | 86.7 ± 1.6 | 22.7 ± 2.3 | 23.7 ± 0.3 | 31.3 ± 4.1 | 105.5 ± 60.7 |
| | C4 | 75.8 ± 1.7 | 24.3 ± 1.6 | 27.3 ± 0.3 | 35.3 ± 2.4 | 64.9 ± 16.6 |
| | PTB | 136.5 ± 11.8 | 46.8 ± 4.6 | 201.0 ± 5.3 | 239.5 ± 6.5 | 122.4 ± 53.7 |
| Calibration: {WikiText2, C4, PTB} Seeds: {0} | WikiText2 | 91.8 ± 4.7 | 20.7 ± 0.9 | 25.5 ± 4.1 | 30.1 ± 5.3 | 126.5 ± 70.3 |
| | C4 | 80.1 ± 2.9 | 23.7 ± 0.4 | 30.2 ± 4.2 | 36.3 ± 3.1 | 133.9 ± 120.0 |
| | PTB | 139.5 ± 10.6 | 41.7 ± 3.8 | 215.5 ± 19.3 | 239.5 ± 6.9 | 165.2 ± 93.7 |

Overall, it can be seen that our method is fairly robust w.r.t. the calibration source, seeds, and number of samples. The only exception is the OPT model, where the sensitivity to the calibration data turns out to be higher. This sensitivity analysis reveals a possible explanation for the results presented in Section 5.2, where our proposed approach consistently outperformed the baselines over all models and tasks, except for the OPT family.

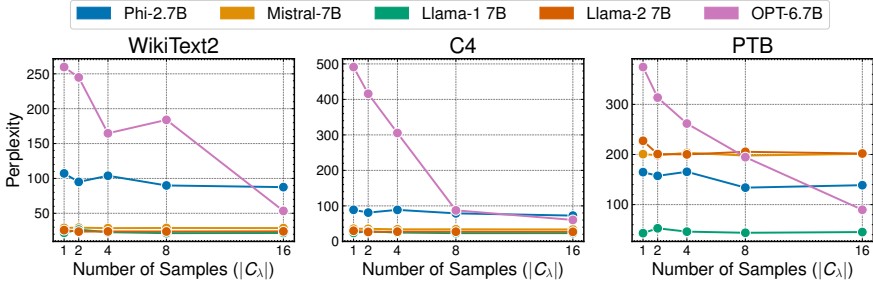

Figure 5: Perplexity over different values of $|C_\lambda|$ (size of the calibration data) when using NEURONAL on the three Language Modeling datasets at 70% sparsity.

# 6 Conclusion and Limitations

In this paper, we have proposed NEURONAL, a new approach to pruning LLMs based on the neuron alignment between sparse and dense activations. The main novelty of our approach is that it exploits information from both the dense and the sparse models while also being adaptive since it is designed to automatically select the best hyperparameters for a given model, pruning algorithm, and target sparsity. Throughout extensive experiments, we showed how our approach outperforms, in most cases, the latest state-of-the-art methods both on Language Modeling datasets and Zero-Shot tasks, with different LLM families and sparsity ratios, while requiring minimal time overhead w.r.t. the base pruning algorithm applied. We also included an extensive sensitivity analysis to show the robustness of our approach to the calibration data, and its capability to select $\lambda$, as well as a runtime comparison with the most recent competitors.

The present version of NEURONAL has two main limitations. The first one derives from the algorithm setup. In fact, NEURONAL requires selecting the size of $\lambda^{\text{set}}$ and $\mathcal{C}_\lambda$, both affecting the computational cost of the forward step. In order to alleviate this limitation and provide a fair comparison w.r.t. the base pruning methods and the top-up baselines, we set $|\mathcal{C}_\lambda| = 8$ (the closest power of 2 to $|C|/|\lambda^{\text{set}}|$), so that NEURONAL incurs only two extra forward passes compared to the base methods, and one more pass than OWL (Yin et al., 2024) and AlphaPruning (Lu et al., 2024). In Appendix B, we analyze the NEURONAL complexity w.r.t. $\lambda^{\text{set}}$ and $\mathcal{C}_\lambda$. The second limitation (which should be noted, however, to be common to *all* top-up pruning algorithms), is the inability to make use of optimized semi-structured sparsity inference implementations (e.g, the NVIDIA N:M sparsity (Pool, 2020)). In fact, for a given sparsity, NEURONAL, as well as OWL (Yin et al., 2024), AlphaPruning (Lu et al., 2024), and DSA (Li et al., 2024), produce customized sparsity constraints for each layer in a block. Therefore, these semi-structured sparsity implementations cannot be employed as they often require continuity in the sparsity (N:M) across *all matrices* in the model.

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

## A  Comparison with Reconstruction Error-Based Pruning

Here, we discuss in detail the main difference between our proposed neuron alignment metric and the well-established reconstruction error (Frantar & Alistarh, 2023). Reconstruction error (Frantar & Alistarh, 2023; Guo et al., 2024; Xu et al., 2024; Zhang et al., 2024) relies on minimizing the L2-norm difference between the outputs of the dense and sparse models at the layer level, following the formula:

$$\min_{\mathcal{M}^\ell} \left\| \mathbf{W}^\ell \mathbf{X}^\ell - (\mathcal{M}^\ell \odot \mathbf{W}^\ell)\mathbf{X}^\ell \right\|_2^2, \tag{9}$$

where $\mathbf{W}^\ell \in \mathbb{R}^{d_{out} \times d_{in}}$ is the weight matrix of layer $\ell$, $\mathbf{X}^\ell \in \mathbb{R}^{d_{in} \times n}$ is the input data to the layer $\ell$, and $\mathcal{M}^\ell \in \{0,1\}^{d_{out} \times d_{in}}$ is the binary pruning mask for the weights of layer $\ell$.

However, in recent decoder-only Transformers (Touvron et al., 2023a; Jiang et al., 2023), each block applies residual connections and intermediate LayerNorms. Hence, preserving the output of a single projection may not be sufficient since it could either be passed to a normalization layer or be summed with the residual stream. Specifically, each module operates as follows:

$$\mathbf{x}_1 = \texttt{LayerNorm}(\mathbf{h}_0)$$
$$\mathbf{a}_1 = \texttt{SelfAttention}(\mathbf{x}_1)$$
$$\mathbf{h}_1 = \mathbf{h}_0 + \mathbf{a}_1$$
$$\mathbf{x}_2 = \texttt{LayerNorm}(\mathbf{h}_1)$$
$$\mathbf{a}_2 = \texttt{MLP}(\mathbf{x}_2)$$
$$\mathbf{h}_2 = \mathbf{h}_1 + \mathbf{a}_2$$

This means that the *input activations to each projection* (i.e., the inputs to $\mathbf{W_Q}$, $\mathbf{W_K}$, $\mathbf{W_V}$, $\mathbf{W_O}$, $\mathbf{W}_{\text{gate}}$, $\mathbf{W}_{\text{up}}$, and $\mathbf{W}_{\text{down}}$) are the real "control points" of information flow. Modifying the weights without preserving these activation patterns can distort the layer's internal behavior.

To address this, NEURONAL directly compares the input activations to each projection in the dense and sparse models. The resulting neuron alignment metric is computed as:

$$\min_{\mathcal{M}} \sum_{x \in C_\lambda} \sum_{b \in \mathcal{B}} \sum_{\ell \in \mathcal{L}} \frac{1}{|A_{\mathcal{D}}^{\ell,b}(x)|} \left\| \frac{A_{\mathcal{D}}^{\ell,b}(x)}{\sum A_{\mathcal{D}}^{\ell,b}(x)} - \frac{A_{\mathcal{S}}^{\ell,b}(x)}{\sum A_{\mathcal{S}}^{\ell,b}(x)} \right\|_2, \tag{10}$$

where $A^\ell(x)$ represents the projection input to layer $\ell$ for input $x$ and $\mathcal{L}$ includes all projection matrices in the Transformer Block $\mathcal{B}_i$.

To conclude, unlike reconstruction-error-based methods, which measure what a layer *produces as output*, our method evaluates what each layer *receives as input*. This distinction leads to a more fine-grained, neuron-level preservation of the behavior of the dense model. Another key difference is given by the fact that the neuron alignment metric produces *one numerical value* for the activations' alignment between the sparse and the dense model, while the reconstruction error provides *separate values* of errors for each sparse layer. This allows us to provide a metric that depicts the global alignment between the sparse and dense models, rather than a local (layer-wise) perspective.

## B  NeuronAL Complexity Analysis

The space complexity of NEURONAL primarily depends on the size of the calibration set $C_\lambda$ and $|\lambda^{\text{set}}|$, i.e., on the number of sparsity schedules to evaluate. For each input $x \in C_\lambda$ and each candidate $\lambda \in \lambda^{\text{set}}$, we perform one forward pass through the dense model and two forward passes through the sparse model (one for the *block* step and one for the *row* step of NEURONAL).

Let $B$ be the number of Transformer blocks, $n$ the input sequence length, and $d$ the hidden dimension. Then, the complexity of one forward pass is $\mathcal{O}(Bn^2d)$. As a result, the overall complexity of NEURONAL becomes:

$$\mathcal{O}\left((1 + 2|\lambda^{\text{set}}|) \cdot |C_\lambda| \cdot Bn^2d\right), \tag{11}$$

From a practical standpoint, we optimize memory usage by processing each block independently and in parallel for the sparse and dense models. Specifically, we collect and compare the input activations to each projection matrix. Once the alignment score is computed for a block, the corresponding activations are discarded to save memory.

As a result, the actual memory complexity at runtime becomes:

$$\mathcal{O}(|C_\lambda| n^2 d), \tag{12}$$

since inference and neuron alignment are performed block-by-block, independently of $|\lambda^{\text{set}}|$ and the number of layers.

### B.1 Complexity NeuronAl vs. OWL

All base pruning methods require a single forward pass over the full calibration set $C$. On the other hand, top-up algorithms, such as OWL (Yin et al., 2024) and AlphaPruning (Lu et al., 2024), require one additional forward pass for computing block information for their non-uniform distribution metric. NEURONAL, instead, performs one forward pass per each $\lambda \in \lambda^{\text{set}}$ over a smaller subset $C_\lambda$. To ensure a comparable cost w.r.t. the top-up baselines which use one additional forward over a calibration set of $|C|$ samples, we set $|C_\lambda| = \frac{|C|}{|\lambda^{\text{set}}|}$, so that the overall number of tokens processed remains roughly the same. As a result, NEURONAL requires $2 \cdot |C_\lambda| \cdot |\lambda^{\text{set}}| = 2 \cdot |C|$ forward passes–which is two times more than base pruning methods, and one time more than OWL and AlphaPruning. The runtime comparisons can be found in Table 11 in the main text.

## C Experimental Setup

**Baselines**. For OWL, we set the hyperparameters to the values that are used mostly in the original paper, hence $M = 5$ and $\lambda = 0.08$; we do the same for AlphaPruning, setting $\epsilon = 0.3$. All these baselines are tested considering each row as a comparison group: in other words, for a given layer, the sparsity $s$ is uniform for each row of each matrix rather than uniform across matrices. This is done for two main reasons: 1) as mentioned earlier, it is established that row-wise pruning on LLMs leads to better performance w.r.t. layer-wise pruning (Sun et al., 2023), and 2) since our approach relies on a row-wise step, for fairness we also use each row (rather than layer) as a comparison group on all the other methods, to appreciate the benefit of our approach. We also test our approach on SparseGPT (Frantar & Alistarh, 2023), using in this case only the *block* step, since SparseGPT relies on a weight reconstruction mechanism that prunes columns first and then adjusts the rows of pruned cells using Hessian information, which makes it unfeasible to apply our *row* step. The results can be found in Tables 28-30. For all the pruning algorithms that use calibration data (i.e., MULTIFLOW, Wanda, and SparseGPT), we use 128 samples from the C4 dataset, as in (Frantar & Alistarh, 2023; Sun et al., 2023; Yin et al., 2024).

**NeuronAl Setup**. Our method takes as input an LLM model, a target sparsity, a scoring-based pruning algorithm, and two sets of $\lambda$ parameters (for the *block* and the *row* steps, respectively). In the experiments, we set $\lambda^{\text{set}} = [0.01, 0.02, 0.03, 0.05, 0.06, 0.07, 0.08, 0.09, 0.1, 0.12, 0.15, 0.20, 0.25]$ for the *block* step, while for the *row* step, we also added 0.0 (in case of no performance improvement). Each value in $\lambda^{\text{set}}$ has been evaluated (as described in Algorithm 3) over a calibration data $C_\lambda$. Since all the base pruning algorithms require a single forward over $C$ (with $C$ containing 128 sequences of 2048 tokens each), while OWL requires a second forward always over $C$, to make the computational runtime similar we set $C_\lambda = 8$ (the closest power of 2 w.r.t. $|C|/|\lambda^{\text{set}}|$). In essence, NEURONAL only requires two forward steps more than the base pruning algorithms, and one forward step more than OWL[5]. All the experiments have been run on NVIDIA A100 GPUs, both with 40 and 80 GB.

---

[5]For both $C$ and $C_\lambda$, we use the same seed (0) for the calibration set, i.e., $C_\lambda$ contains the first 8 elements of $C$.

# D  Additional experiments

Here we include the results of the experiments that, due to space limits, we could not include in the main text. Specifically, we report: the results of NEURONAL over a more recent model, namely LLama-3 8B; the results on the Language Modeling datasets at 60% and 80% sparsity and the full results on the Zero-Shot tasks at 60%, 70%, and 80% sparsity; the results for the Zero-Shot tasks with Magnitude pruning; the full results of Zero-Shot tasks over LLama 13B and 70B at 70% sparsity; the results of NEURONAL (*block*-only) applied to SparseGPT (Frantar & Alistarh, 2023).

## D.1  Results on LLama-3 8B

In this section, we include further experiments over a more recent model: LLama-3 8B. We stick to the same experimental setup of Appendix C and compare NEURONAL to the same baselines of the main text. Specifically, we report the Language Modeling results in Table 15 and Zero-Shot results in Table 16. As we saw in the main text for the case of LLama-2 70B, also in this case, our method offers competitive results (although it does not always result in being the best algorithm in all cases) over the Language Modeling

Table 15: Perplexity on the three Language Modeling datasets computed over LLama-3 8B) for four different top-up algorithms (Uniform, DSnoT, OWL, and NEURONAL) on three pruning algorithms (Magnitude, MULTIFLOW, and Wanda) at 70% sparsity.

| Algorithm | Top-Up | 60% | | | 70% | | | 80% | | |
|---|---|---|---|---|---|---|---|---|---|---|
| | | WikiText2 | C4 | PTB | WikiText2 | C4 | PTB | WikiText2 | C4 | PTB |
| Dense | | 6.1 | 9.4 | 11.2 | | | | | | |
| MULTIFLOW | Uniform | 35.4 | 41.9 | 51.3 | 151.3 | 168.1 | 156.1 | 1397.5 | 1058.5 | 2125.0 |
| | DSnoT | 23.8 | 32.9 | 40.1 | **119.6** | 152.0 | **148.8** | 1497.0 | 1066.1 | 1775.2 |
| | OWL | 27.7 | 32.0 | 42.0 | 126.7 | 138.0 | 152.9 | **891.6** | **620.7** | **716.5** |
| | AlphaPruning | **22.6** | 31.5 | 40.4 | 151.0 | 170.6 | 218.0 | 1309.1 | 1125.3 | 1800.9 |
| | NEURONAL | 23.7 | **30.2** | **38.7** | 132.9 | **129.4** | 187.9 | 5533.5 | 3975.1 | 6395.9 |
| Wanda | Uniform | 23.6 | 35.3 | 41.6 | 125.2 | 163.0 | 147.6 | 1051.5 | 824.1 | 1029.9 |
| | DSnoT | 21.4 | 30.6 | 36.4 | 121.8 | 158.0 | 152.5 | 1149.9 | 717.9 | 1013.6 |
| | OWL | 18.6 | 26.4 | 31.7 | **92.1** | 124.8 | **132.3** | 721.3 | 601.1 | 913.0 |
| | AlphaPruning | 19.2 | 28.4 | 33.1 | 115.7 | 157.1 | 167.7 | 1110.1 | 1079.0 | 1144.8 |
| | NEURONAL | **18.4** | **25.6** | **30.4** | 93.7 | **98.1** | 159.6 | 777.2 | 604.2 | 1012.6 |

datasets. However, it is by a large margin the most performing algorithm, on average, across the Zero-Shot tasks, as clearly shown in Table 16. Overall, the points of strength of NEURONAL stated in the main text also apply to the most recent LLM models.

Table 16: Accuracy on the seven Zero-Shot Tasks, computed over LLama-3 8B for four different top-up pruning algorithms (DSnoT, OWL, AlphaPruning, and NEURONAL) on two pruning algorithms (MULTIFLOW and Wanda) at 70% and 80% sparsity. "Average" indicates the mean accuracy across tasks. The rows corresponding to the pruning algorithms refer to the uniform distribution.

| Sparsity | Algorithm | RTE | WinoGrande | BoolQ | HellaSwag | ARC-e | ARC-c | OBQA | Average |
|---|---|---|---|---|---|---|---|---|---|
| Dense | | 69.68 | 72.77 | 81.35 | 60.19 | 80.09 | 50.43 | 34.8 | 64.19 |
| 70% | MULTIFLOW | 52.71 | 48.86 | 51.90 | 27.55 | 31.19 | 18.26 | 12.20 | 34.67 |
| | w. DSnoT | 52.71 | 48.46 | 50.09 | 27.31 | 31.36 | 17.15 | 13.40 | 34.35 |
| | w. OWL | 52.71 | 51.62 | 62.11 | 27.84 | 34.05 | **17.92** | 12.20 | 36.92 |
| | AlphaPruning | 52.71 | 50.99 | 61.19 | 27.82 | 32.03 | 17.24 | 12.80 | 36.40 |
| | w. NEURONAL | 52.71 | **53.12** | **62.17** | **29.21** | **37.12** | 17.83 | **15.20** | **38.19** |
| | Wanda | 52.71 | 49.01 | 51.38 | 27.27 | 32.53 | 17.58 | 12.00 | 34.64 |
| | w. DSnoT | 52.71 | 48.38 | 51.50 | 27.19 | 31.06 | 17.58 | 13.60 | 34.57 |
| | w. OWL | 52.71 | 48.86 | 61.59 | 28.29 | 35.69 | 17.41 | 12.80 | 36.76 |
| | AlphaPruning | 52.71 | 50.51 | 53.58 | 27.69 | 34.64 | 17.15 | 13.20 | 35.64 |
| | w. NEURONAL | 52.71 | **54.06** | **62.20** | **31.27** | **41.71** | **20.05** | **16.20** | **39.74** |
| 80% | MULTIFLOW | 52.71 | **50.28** | 37.86 | 26.46 | 27.95 | 19.54 | 11.00 | 32.26 |
| | w. DSnoT | 52.71 | 46.57 | 37.95 | 26.36 | 28.03 | **20.48** | 11.80 | 31.99 |
| | w. OWL | 52.71 | 48.62 | 40.76 | 26.38 | 28.24 | 20.22 | 13.00 | 32.85 |
| | AlphaPruning | 52.71 | 48.46 | 49.85 | 26.42 | 27.19 | 19.20 | **14.20** | 34.00 |
| | w. NEURONAL | 52.35 | 49.01 | **62.14** | **26.54** | **28.41** | 19.45 | 13.60 | **35.93** |
| | Wanda | 52.71 | 47.99 | 37.83 | **26.68** | 27.53 | 19.11 | 12.40 | 32.04 |
| | w. DSnoT | 52.71 | 47.51 | 37.80 | 26.21 | 28.24 | **21.25** | 12.60 | 32.33 |
| | w. OWL | 52.71 | 47.83 | 38.20 | 26.63 | **28.58** | 19.54 | 13.60 | 32.44 |
| | AlphaPruning | **53.07** | 49.41 | 48.04 | 26.39 | 27.15 | 19.88 | **13.60** | 33.93 |
| | w. NEURONAL | 52.71 | **49.57** | **60.31** | 26.65 | 28.41 | 18.69 | 13.40 | **35.68** |

## D.2 Language Modeling at 60% and 80% Sparsity

In Table 17-18, we report the results of NeuronAl over the 3 Language Modeling datasets (WikiText2, C4, and PTB) with the five different LLMs considered in the main text, for 60% and 80% sparsity. In the first case, our approach turns out to be the best one in 23 out of 45 cases, while for 80% sparsity in 20 out of 45, while is second best in 15 cases. It is interesting to notice how at medium sparsity (60%) all the top-up algorithms, including ours, provide similar results, while the improvement provided by NeuronAl at 80% (w.r.t. the top-up competitors) in some cases reaches a factor of 2-3x (e.g., with LLama-1 7B for MULTIFLOW and Wanda).

Table 17: Perplexity on the three Language Modeling datasets computed over five different LLMs for four different top-up algorithms (Uniform, DSnoT, OWL, and NeuronAl) on three pruning algorithms (Magnitude, MULTIFLOW, and Wanda) at 60% sparsity.

| Algorithm | Top-Up | Phi-2.7B | | | LLama-1 7B | | | LLama-2 7B | | | Mistral-7B | | | OPT 6.7B | | |
|---|---|---|---|---|---|---|---|---|---|---|---|---|---|---|---|---|
| | | WikiText2 | C4 | PTB | WikiText2 | C4 | PTB | WikiText2 | C4 | PTB | WikiText2 | C4 | PTB | WikiText2 | C4 | PTB |
| Dense | | 9.7 | 14.1 | 18.2 | 5.7 | 7.3 | 10.1 | 5.5 | 7.3 | 32.9 | 5.3 | 8.4 | 36.6 | 10.9 | 12.7 | 15.8 |
| Magnitude | Uniform | 51.3 | 45.9 | 66.9 | 152.4 | 159.8 | 3.02e3 | 6.89e3 | 4.27e4 | 1.71e6 | 19.6 | 24.4 | 189.3 | 9.49e3 | 6.20e3 | 6.76e3 |
| | DSnoT | 55.9 | 48.9 | **64.1** | 131.6 | 114.7 | 1.46e3 | 3.68e3 | 6.78e4 | 7.30e6 | 15.3 | 19.6 | 146.2 | 8.08e3 | 6.06e3 | 6.95e3 |
| | OWL | **46.5** | **42.2** | 65.7 | **50.5** | **62.9** | 249.4 | 810.9 | 1.94e4 | 2.30e6 | **12.0** | **15.8** | 169.1 | 6.81e3 | 3.67e3 | 4.07e3 |
| | AlphaPruning | 268.7 | 254.2 | 1.13e3 | 61.6 | 82.2 | 153.6 | **37.1** | **49.4** | 5.43e5 | 13.1 | 18.2 | 185.4 | 5.07e3 | 2.62e3 | 3.03e3 |
| | NeuronAl | 48.5 | 44.8 | 65.1 | 55.8 | 63.6 | 153.1 | 52.4 | 79.3 | 2.48e4 | 12.5 | 17.0 | 139.0 | 1.20e3 | 547.6 | 932.6 |
| MULTIFLOW | Uniform | 25.4 | 28.3 | 54.7 | 11.6 | 13.9 | 26.0 | 11.0 | 13.7 | 166.9 | 167.7 | 341.0 | 884.3 | 16.3 | 19.7 | 26.8 |
| | DSnoT | 37.2 | 42.1 | 50.2 | 10.1 | 12.7 | 18.2 | 10.5 | 13.4 | 137.6 | **10.9** | **14.8** | **86.0** | **15.9** | 19.2 | **25.3** |
| | OWL | 23.8 | **26.7** | 49.8 | 10.6 | 12.9 | 19.6 | 10.1 | 12.7 | 106.0 | 84.1 | 123.0 | 644.6 | 16.1 | **18.5** | 25.5 |
| | AlphaPruning | 434.1 | 401.2 | 1.82e3 | 11.6 | 14.1 | 19.9 | 10.9 | 13.8 | 83.4 | 89.1 | 116.4 | 454.6 | 30.9 | 25.4 | 45.3 |
| | NeuronAl | **23.6** | 27.0 | **42.0** | **9.9** | **12.2** | **17.7** | **9.7** | **12.1** | **72.2** | 112.4 | 168.0 | 806.9 | 16.6 | 19.9 | 27.3 |
| Wanda | Uniform | 225.8 | 29.3 | 48.9 | 10.7 | 13.7 | 24.0 | 10.8 | 14.0 | 122.2 | 11.3 | 15.9 | 101.6 | **15.2** | **17.9** | **23.7** |
| | DSnoT | 32.2 | 38.0 | 50.6 | 10.4 | 13.2 | 20.8 | 10.8 | 14.1 | 109.6 | 11.4 | 15.9 | 96.8 | 15.8 | 19.1 | 24.6 |
| | OWL | **24.8** | 28.2 | 48.6 | **9.4** | **11.8** | 18.5 | **9.2** | **11.9** | 75.1 | 10.3 | 14.5 | 84.5 | 15.7 | 17.8 | 24.5 |
| | AlphaPruning | 165.3 | 166.5 | 669.7 | 10.1 | 12.7 | 17.7 | 9.8 | 12.6 | 69.3 | 10.8 | 15.3 | 93.5 | 27.9 | 24.3 | 41.2 |
| | NeuronAl | 25.3 | **27.1** | **41.8** | 9.6 | 12.0 | **17.4** | 9.4 | 12.0 | **64.3** | **9.9** | **13.8** | **81.9** | 16.3 | 19.1 | 25.2 |

Table 18: Perplexity on the three Language Modeling datasets computed over five different LLMs for four different top-up algorithms (Uniform, DSnoT, OWL, and NeuronAl) on three pruning algorithms (Magnitude, MULTIFLOW, and Wanda) at 80% sparsity.

| Algorithm | Top-Up | Phi-2.7B | | | LLama-1 7B | | | LLama-2 7B | | | Mistral-7B | | | OPT 6.7B | | |
|---|---|---|---|---|---|---|---|---|---|---|---|---|---|---|---|---|
| | | WikiText2 | C4 | PTB | WikiText2 | C4 | PTB | WikiText2 | C4 | PTB | WikiText2 | C4 | PTB | WikiText2 | C4 | PTB |
| Dense | | 9.7 | 14.1 | 18.2 | 5.7 | 7.3 | 10.1 | 5.5 | 7.3 | 32.9 | 5.3 | 8.4 | 36.6 | 10.9 | 12.7 | 15.8 |
| Magnitude | Uniform | 1.53e4 | 1.79e4 | 3.20e4 | 1.13e5 | 1.14e5 | 1.40e5 | 5.58e4 | 5.26e4 | 8.98e4 | 2.48e4 | 3.12e4 | 7.98e3 | 4.29e4 | 2.13e4 | 2.21e4 |
| | DSnoT | 1.99e4 | 2.07e4 | 1.84e4 | **3.40e4** | **3.42e4** | **7.20e4** | 2.36e6 | 1.80e6 | 3.02e6 | 1.33e4 | 8.03e3 | 5.80e3 | 1.81e4 | **1.19e4** | **1.44e4** |
| | OWL | **6.63e3** | **5.60e3** | **8.39e3** | 1.69e5 | 1.59e5 | 1.34e5 | **2.69e4** | **1.79e4** | **5.79e4** | 9.61e3 | 8.50e3 | 5.79e3 | 3.32e4 | 1.78e4 | 2.16e4 |
| | AlphaPruning | 8.54e4 | 9.37e4 | 7.83e4 | 5.78e4 | 4.54e4 | 9.22e4 | 4.83e4 | 2.46e4 | 5.09e4 | 3.03e3 | 2.74e3 | 3.30e3 | 2.52e4 | 1.31e4 | 1.32e4 |
| | NeuronAl | 9.34e3 | 8.36e3 | 1.14e4 | 8.23e4 | 8.57e4 | 1.08e5 | 3.64e4 | 3.15e4 | 2.99e4 | 1.42e4 | 7.53e2 | 4.45e3 | 3.42e4 | 2.27e4 | 1.84e4 |
| MULTIFLOW | Uniform | 2.53e4 | 1.28e4 | 2.59e4 | 4.83e3 | 2.31e3 | 9.81e3 | 2.04e3 | 1.46e3 | 3.88e3 | 4.29e3 | 2.98e3 | 3.81e3 | 4.42e3 | 2.38e3 | 3.28e3 |
| | DSnoT | 8.50e3 | 3.92e3 | 1.23e4 | 3.70e3 | 2.65e3 | 8.26e3 | 1.72e3 | 1.54e3 | 3.44e3 | 327.0 | 270.0 | 752.5 | 1.16e4 | 9.72e3 | 1.18e4 |
| | OWL | 255.2 | 2.80e3 | 1.36e4 | 926.5 | 563.1 | 1.78e3 | 544.2 | **414.3** | 2.82e3 | 3.35e3 | 2.21e3 | 3.56e3 | 1.35e4 | 1.11e4 | 1.51e4 |
| | AlphaPruning | 2.12e4 | 1.27e4 | 2.04e4 | 934.0 | 841.6 | 1.55e3 | 899.1 | 670.9 | 2.60e3 | 3.46e3 | 3.69e3 | 6.05e3 | 4.21e3 | 2.92e3 | 3.48e3 |
| | NeuronAl | **2.34e3** | 992.4 | **4.22e3** | **259.8** | **209.2** | **613.8** | **378.5** | 456.8 | **2.09e3** | **1.02e3** | **719.3** | **1.56e3** | **1.29e3** | **721.9** | **1.35e3** |
| Wanda | Uniform | 2.05e4 | 1.24e4 | 3.14e4 | 5.22e3 | 3.97e3 | 1.00e4 | 4.93e3 | 3.12e3 | 5.29e3 | 330.9 | 277.7 | 783.7 | 4.26e3 | 2.35e3 | 2.73e3 |
| | DSnoT | 1.53e4 | 6.86e3 | 1.40e4 | 3.71e3 | 3.08e3 | 7.79e3 | 5.20e3 | 4.44e3 | 6.69e3 | 346.5 | 277.3 | 758.4 | 7.75e3 | 6.16e3 | 7.78e3 |
| | OWL | **2.55e3** | **1.21e3** | 7.06e3 | 986.5 | 654.5 | 2.00e3 | 663.0 | 486.2 | 2.28e3 | 206.3 | 187.8 | **603.9** | 1.32e4 | 1.06e4 | 1.42e4 |
| | AlphaPruning | 4.31e4 | 3.33e4 | 2.66e4 | 768.4 | 654.9 | 1.29e3 | 982.1 | 670.0 | **2.18e3** | **204.3** | 182.6 | 774.2 | 5.61e3 | 4.97e3 | 5.21e3 |
| | NeuronAl | 2.50e3 | 1.59e3 | **4.03e3** | **302.8** | **272.2** | **783.8** | **557.2** | 660.2 | 2404.1 | 249.5 | **177.6** | 783.9 | **1.04e3** | 632.8 | **1.13e3** |

### D.3 Zero-Shot at 60%, 70%, and 80% Sparsity

In Tables 19-21, we report the results at 60%, 70%, and 80% sparsity of NEURONAL over Zero-Shot tasks with the five different LLMs tested in the main text. In particular, we report the detailed results for each task, while in the main text, we only report the average results across the seven tasks.

Table 19: Accuracy on the seven Zero-Shot Tasks, computed over five different LLMs for three different top-up pruning algorithms (DSnoT, OWL, and NEURONAL) on two pruning algorithms (MULTIFLOW and Wanda) at 60% sparsity. "Average" indicates the mean accuracy across tasks. The rows corresponding to the pruning algorithms refer to the uniform distribution.

| Model | Algorithm | RTE | WinoGrande | BoolQ | HellaSwag | ARC-e | ARC-c | OBQA | Average |
|---|---|---|---|---|---|---|---|---|---|
| | MULTIFLOW | 62.09 | **67.64** | 63.15 | 42.04 | **71.76** | **39.51** | 27.2 | 53.34 |
| | w. DSnoT | 63.18 | 66.77 | 43.49 | 40.84 | 66.62 | 35.41 | 26.0 | 48.90 |
| | w. OWL | **64.62** | 67.17 | 60.15 | 41.78 | 70.96 | 37.8 | **28.8** | 53.04 |
| | AlphaPruning | 62.45 | 52.96 | 50.83 | 29.03 | 48.27 | 22.18 | 16.0 | 40.25 |
| | w. NEURONAL | 62.82 | 67.4 | 65.87 | 42.96 | 70.88 | 38.65 | 27.8 | **53.77** |
| Phi-2.7B | Wanda | 63.54 | 64.8 | 69.08 | 40.16 | 68.64 | 34.9 | 25.4 | **52.36** |
| | w. DSnoT | 62.45 | 64.33 | 59.17 | 39.25 | 64.18 | 33.53 | 22.6 | 49.36 |
| | w. OWL | **64.62** | 64.33 | 64.83 | 39.80 | **67.63** | **34.98** | **24.2** | 51.48 |
| | AlphaPruning | 64.26 | 55.33 | 62.39 | 31.07 | 47.9 | 23.81 | 17.2 | 43.14 |
| | w. NEURONAL | 64.26 | 66.38 | 67.55 | 40.63 | 66.5 | 34.39 | 24.6 | 52.04 |
| | MULTIFLOW | **57.04** | 62.51 | 67.19 | 45.31 | 59.55 | 30.97 | 24.6 | 49.60 |
| | w. DSnoT | 49.46 | 63.06 | 68.32 | 44.75 | **63.80** | 31.23 | 26.4 | 49.57 |
| | w. OWL | 54.15 | **63.46** | 66.54 | 46.53 | 60.65 | 31.14 | 25.4 | 49.70 |
| | AlphaPruning | 50.9 | 65.43 | 67.65 | 46.11 | 58.08 | 31.57 | 24.2 | 49.13 |
| | w. NEURONAL | 50.9 | 63.54 | 68.35 | 47.91 | 63.89 | 32.17 | 27.2 | **50.57** |
| LLama-1 7B | Wanda | **59.57** | 62.67 | 68.81 | 43.64 | 62.84 | 30.38 | 24.8 | 50.39 |
| | w. DSnoT | 51.62 | 61.64 | 67.37 | 43.39 | 63.89 | 30.55 | 25.4 | 49.12 |
| | w. OWL | 55.60 | **64.17** | **70.61** | **46.63** | 62.96 | **31.74** | 24.8 | 50.93 |
| | AlphaPruning | 59.57 | 65.27 | 68.81 | 46.09 | 60.4 | 32.85 | 24.6 | 51.08 |
| | w. NEURONAL | 58.48 | 63.61 | 70.55 | 46.53 | 63.8 | 30.89 | 26.0 | **51.41** |
| | MULTIFLOW | **57.04** | 61.96 | 64.80 | 43.39 | 60.44 | 29.1 | 24.6 | 48.76 |
| | w. DSnoT | 54.15 | 63.77 | 63.91 | 43.42 | 66.25 | 31.83 | 24.0 | 49.62 |
| | w. OWL | 54.87 | 62.75 | 65.14 | 45.20 | 62.58 | 29.52 | 23.8 | 49.12 |
| | AlphaPruning | 52.71 | 64.96 | 64.86 | 44.63 | 61.32 | 30.8 | 23.4 | 48.95 |
| | w. NEURONAL | 53.07 | 65.27 | 69.27 | 46.85 | 66.62 | 31.31 | 25.2 | **51.08** |
| LLama-2 7B | Wanda | **54.15** | 64.48 | 65.44 | 43.85 | 65.19 | 30.46 | 26.2 | 49.97 |
| | w. DSnoT | 53.79 | 64.09 | 64.83 | 42.39 | 63.89 | 30.03 | 22.8 | 48.83 |
| | w. OWL | 53.79 | **66.61** | 66.76 | 46.63 | **67.63** | 32.34 | **28.0** | 51.68 |
| | AlphaPruning | 54.15 | 67.4 | 66.67 | 46.17 | 64.98 | 33.28 | 25.8 | 51.21 |
| | w. NEURONAL | 52.71 | 66.77 | 71.99 | 46.85 | 66.33 | 32.08 | 27.2 | **51.99** |
| | MULTIFLOW | 51.62 | 49.88 | 39.17 | 27.49 | 29.67 | 18.77 | 11.0 | 32.51 |
| | w. DSnoT | **54.87** | **66.61** | **70.86** | **45.93** | **68.27** | **32.94** | **24.0** | **51.93** |
| | w. OWL | 53.07 | 50.12 | 46.33 | 28.29 | 32.58 | 19.28 | 13.2 | 34.70 |
| | AlphaPruning | 53.79 | 51.93 | 43.21 | 27.78 | 35.52 | 19.37 | 13.2 | 34.97 |
| | w. NEURONAL | 52.35 | 51.93 | 43.33 | 28.08 | 32.28 | 18.52 | 12.0 | 34.07 |
| Mistral-7B | Wanda | 54.87 | 66.06 | 71.13 | 44.48 | 67.05 | 32.00 | 21.60 | 51.03 |
| | w. DSnoT | 54.15 | 65.59 | 70.43 | 44.5 | 66.88 | 31.40 | 20.60 | 50.51 |
| | w. OWL | **57.04** | 67.17 | **73.85** | 45.66 | **67.89** | 32.59 | 23.20 | **52.49** |
| | AlphaPruning | 57.4 | 66.54 | 64.98 | 44.02 | 65.32 | 30.03 | 20.8 | 49.87 |
| | w. NEURONAL | 557.04 | 66.06 | 70.34 | 46.37 | 68.6 | 33.02 | 23.2 | 52.09 |
| | MULTIFLOW | 52.71 | 58.25 | 62.69 | 42.24 | 56.52 | 25.68 | 22.20 | 45.76 |
| | w. DSnoT | **53.07** | 58.48 | 62.57 | 42.13 | **57.79** | 23.98 | 22.20 | 45.75 |
| | w. OWL | **53.07** | 57.46 | **63.21** | **42.98** | 56.44 | **25.68** | **24.20** | **46.15** |
| | AlphaPruning | 58.12 | 58.64 | 62.29 | 40.83 | 51.68 | 24.49 | 19.0 | 45.01 |
| | w. NEURONAL | 52.71 | 58.33 | 62.32 | 42.19 | 56.4 | 25.0 | 21.6 | 45.51 |
| OPT-6.7B | Wanda | 52.71 | 59.67 | 62.29 | **42.80** | 58.00 | **25.60** | 22.6 | **46.24** |
| | w. DSnoT | 52.71 | 58.17 | 62.11 | 41.99 | 57.41 | 25.43 | 22.4 | 45.75 |
| | w. OWL | 52.71 | 58.72 | **62.69** | 42.14 | **58.33** | 25.17 | 22.6 | 46.05 |
| | AlphaPruning | 55.23 | 58.09 | 62.17 | 39.71 | 51.85 | 24.83 | 19.6 | 44.50 |
| | w. NEURONAL | 52.71 | 61.25 | 62.39 | 41.95 | 57.28 | 25.09 | 22.0 | 46.10 |

Table 20: Accuracy on the seven Zero-Shot Tasks, computed over five different LLMs for three different top-up pruning algorithms (DSnoT, OWL, and NEURONAL) on two pruning algorithms (MULTIFLOW and Wanda) at 70% sparsity. "Average" indicates the mean accuracy across tasks. The rows corresponding to the pruning algorithms refer to the uniform distribution.

| Model | Algorithm | RTE | WinoGrande | BoolQ | HellaSwag | ARC-e | ARC-c | OBQA | Average |
|---|---|---|---|---|---|---|---|---|---|
| | MULTIFLOW | **53.79** | 50.43 | 53.49 | 28.43 | 46.59 | 20.48 | 15.6 | 38.40 |
| | w. DSnoT | 52.71 | 54.06 | 54.92 | 28.38 | 44.15 | 21.5 | 15.6 | 38.76 |
| | w. OWL | 52.35 | 55.33 | 61.99 | 30.31 | **54.59** | 24.23 | **18.4** | 42.46 |
| | w. AlphaPruning | 47.29 | 51.3 | 37.71 | 25.85 | 26.64 | 20.9 | 13.0 | 31.81 |
| | w. NEURONAL | 53.07 | 58.64 | 62.17 | 33.45 | 52.19 | 25.34 | 17.4 | **43.18** |
| Phi-2.7B | Wanda | 52.35 | 53.2 | 62.14 | 28.31 | 44.87 | 20.99 | 17.4 | 39.89 |
| | w. DSnoT | 52.35 | 51.54 | 60.98 | 28.33 | 41.62 | 21.08 | 15.4 | 38.76 |
| | w. OWL | **52.71** | 53.59 | 62.05 | 30.09 | 48.61 | 22.53 | **18.8** | 41.20 |
| | w. AlphaPruning | 47.29 | 49.88 | 60.86 | 25.52 | 25.55 | 22.01 | 14.0 | 35.02 |
| | w. NEURONAL | 52.71 | 56.35 | 62.17 | 32.47 | 49.62 | 22.95 | 16.8 | **41.87** |
| | MULTIFLOW | 55.96 | 52.57 | 61.96 | 29.77 | 34.64 | 19.45 | 15.2 | 38.51 |
| | w. DSnoT | 54.15 | 50.43 | 59.33 | 29.33 | 36.45 | 19.28 | 13.6 | 37.51 |
| | w. OWL | 52.35 | 58.64 | 62.63 | 36.74 | 47.43 | 26.62 | 18.2 | 43.23 |
| | w. AlphaPruning | 55.6 | 63.54 | 64.4 | 36.37 | 43.22 | 26.79 | 18.6 | 44.07 |
| | w. NEURONAL | 57.76 | 61.72 | 63.3 | 38.19 | 50.88 | 26.88 | 22.8 | **45.93** |
| LLama-1 7B | Wanda | 55.23 | 52.8 | 57.46 | 28.84 | 32.2 | 18.0 | 13.8 | 36.9 |
| | w. DSnoT | 54.15 | 51.22 | 54.56 | 28.97 | 33.08 | 18.26 | 13.6 | 36.26 |
| | w. OWL | **58.48** | 58.56 | 62.60 | 34.74 | 47.35 | 24.06 | 17.4 | 43.31 |
| | w. AlphaPruning | 56.68 | 63.14 | 63.85 | 35.78 | 44.82 | 26.71 | 17.6 | 44.08 |
| | w. NEURONAL | 55.96 | 59.27 | 63.12 | 36.94 | 50.0 | 26.11 | 20.6 | **44.57** |
| | MULTIFLOW | 52.71 | 50.99 | 62.05 | 28.52 | 33.04 | 17.92 | 13.6 | 36.98 |
| | w. DSnoT | 52.71 | 50.99 | 59.72 | 27.92 | 32.58 | 16.81 | 13.0 | 36.25 |
| | w. OWL | 52.71 | 56.12 | 62.05 | 32.40 | 42.42 | 19.88 | 18.6 | 40.6 |
| | w. AlphaPruning | 52.71 | 58.64 | 62.2 | 35.17 | 41.33 | 22.87 | 17.6 | 41.5 |
| | w. NEURONAL | 53.43 | 58.09 | 62.35 | 35.26 | 48.32 | 22.44 | 20.2 | **42.87** |
| LLama-2 7B | Wanda | 52.71 | 48.46 | 49.94 | 28.09 | 30.39 | 19.2 | 11.8 | 34.37 |
| | w. DSnoT | 52.71 | 50.36 | 47.77 | 27.67 | 30.6 | 17.32 | 12.2 | 34.09 |
| | w. OWL | 52.71 | 55.96 | 62.11 | 31.86 | 43.73 | 20.65 | 17.0 | 40.57 |
| | w. AlphaPruning | 52.71 | 61.33 | 62.2 | 34.82 | 43.43 | 22.1 | 17.6 | 42.03 |
| | w. NEURONAL | 53.07 | 57.85 | 63.27 | 35.42 | 49.62 | 22.44 | 20.0 | **43.1** |
| | MULTIFLOW | 49.82 | 50.75 | 41.19 | 26.45 | 26.64 | **21.84** | 12.6 | 32.76 |
| | w. DSnoT | 52.71 | **52.57** | **62.42** | **29.51** | **36.66** | 18.94 | 12.0 | **37.83** |
| | w. OWL | **53.79** | 49.17 | 38.90 | 26.77 | 27.78 | 19.20 | 12.8 | 32.63 |
| | w. AlphaPruning | 52.35 | 48.07 | 37.95 | 27.01 | 28.28 | 18.17 | 11.8 | 31.95 |
| | w. NEURONAL | 52.71 | 50.75 | 38.29 | 27.16 | 28.75 | 17.75 | 13.0 | 32.63 |
| Mistral-7B | Wanda | 52.71 | 51.62 | 59.79 | 28.86 | 34.18 | 18.17 | 12.6 | 36.85 |
| | w. DSnoT | 52.71 | 50.28 | 58.62 | 28.51 | 33.54 | 18.86 | 13.2 | 36.53 |
| | w. OWL | 52.71 | 53.91 | **62.20** | 30.95 | 39.39 | 18.60 | 13.6 | 38.77 |
| | w. AlphaPruning | 52.71 | 56.27 | 62.2 | 31.47 | 38.55 | 18.52 | 13.6 | 39.05 |
| | w. NEURONAL | 52.71 | 60.62 | 62.17 | 34.8 | 44.28 | 20.31 | 16.0 | **41.56** |
| | MULTIFLOW | **53.79** | 49.72 | 43.0 | 26.48 | 30.51 | 20.05 | 13.8 | 33.91 |
| | w. DSnoT | **53.79** | 49.01 | 61.1 | 27.01 | 32.87 | 18.34 | 12.2 | 36.33 |
| | w. OWL | 48.74 | 48.62 | 61.56 | 27.18 | 35.69 | 16.47 | 11.6 | 35.69 |
| | AlphaPruning | 46.93 | 51.78 | 62.17 | 31.48 | 36.49 | 22.18 | 13.8 | 37.83 |
| | w. NEURONAL | 50.54 | 50.99 | 62.17 | 31.3 | 40.74 | 22.27 | 17.4 | **39.34** |
| OPT-6.7B | Wanda | 52.71 | 49.72 | 60.03 | 26.91 | 35.86 | 17.75 | 11.2 | 36.31 |
| | w. DSnoT | 52.71 | 49.57 | 60.61 | 26.91 | 35.06 | 17.58 | 12.0 | 36.35 |
| | w. OWL | **53.79** | 51.22 | 61.87 | 29.53 | **42.3** | 18.09 | **14.6** | 38.77 |
| | AlphaPruning | 51.62 | 51.7 | 62.17 | 33.13 | 40.19 | 22.7 | 15.2 | 39.53 |
| | w. NEURONAL | 50.90 | 51.07 | 62.17 | 30.54 | 40.78 | 21.16 | 15.4 | **38.86** |

Table 21: Accuracy on the seven Zero-Shot Tasks, computed over five different LLMs for three different top-up pruning algorithms (DSnoT, OWL, and NEURONAL) on two pruning algorithms (MULTIFLOW and Wanda) at 80% sparsity. "Average" indicates the mean accuracy across tasks. The rows corresponding to the pruning algorithms refer to the uniform distribution.

| Model | Algorithm | RTE | WinoGrande | BoolQ | HellaSwag | ARC-e | ARC-c | OBQA | Average |
|---|---|---|---|---|---|---|---|---|---|
| Phi-2.7B | MULTIFLOW | 52.71 | 50.12 | 51.04 | 26.15 | 27.44 | 19.37 | 13.4 | 34.32 |
| | w. DSnoT | 50.18 | 49.49 | 37.83 | 26.32 | 27.82 | **19.97** | 13.8 | 32.2 |
| | w. OWL | **53.07** | **51.22** | **56.76** | 26.30 | 30.85 | 18.52 | **15.2** | **35.99** |
| | AlphaPruning | 44.77 | 48.78 | 37.83 | 25.74 | 26.18 | 20.31 | 11.4 | 30.72 |
| | w. NEURONAL | 52.71 | 50.59 | 43.0 | 26.66 | 33.88 | 21.16 | 16.0 | 34.86 |
| | Wanda | **53.07** | 49.25 | 62.17 | 25.93 | 27.44 | 20.99 | 14.6 | 36.21 |
| | w. DSnoT | **53.07** | 50.99 | 38.04 | 26.17 | 27.10 | 20.73 | 13.0 | 32.73 |
| | w. OWL | 52.35 | **51.22** | 60.21 | 26.51 | 29.88 | 20.05 | 13.6 | 36.26 |
| | AlphaPruning | 43.32 | 49.8 | 38.29 | 25.94 | 25.67 | 21.16 | 11.0 | 30.74 |
| | w. NEURONAL | 52.71 | 50.2 | 62.17 | 26.96 | 32.32 | 20.56 | 16.6 | **37.36** |
| LLama-1 7B | MULTIFLOW | 47.29 | **50.91** | 40.03 | 26.17 | 26.77 | **21.16** | 12.4 | 32.1 |
| | w. DSnoT | 46.57 | 50.43 | 37.83 | 26.02 | 27.06 | 20.05 | 12.2 | 31.45 |
| | w. OWL | 50.18 | 50.04 | 45.47 | 26.74 | 27.65 | 20.48 | 12.2 | 33.25 |
| | AlphaPruning | 56.68 | 49.96 | 62.17 | 26.44 | 27.23 | 21.25 | 11.2 | 36.42 |
| | w. NEURONAL | 54.51 | 51.14 | 62.2 | 28.82 | 32.74 | 19.45 | 14.4 | **37.61** |
| | Wanda | 47.29 | 49.88 | 37.83 | 26.34 | 26.47 | 20.99 | 12.8 | 31.66 |
| | w. DSnoT | 46.93 | **50.36** | 37.83 | 26.03 | 26.56 | **21.33** | **13.4** | 31.78 |
| | w. OWL | 47.29 | 49.88 | 37.83 | 26.67 | 27.19 | 19.54 | 11.6 | 31.43 |
| | AlphaPruning | 52.35 | 49.17 | 61.19 | 26.58 | 26.77 | 20.73 | 10.6 | 35.34 |
| | w. NEURONAL | 52.71 | 49.57 | 60.03 | 28.21 | 30.85 | 19.62 | 13.2 | **36.31** |
| LLama-2 7B | MULTIFLOW | **53.43** | 48.86 | 37.83 | 26.35 | 27.48 | **21.25** | 13.2 | 32.63 |
| | w. DSnoT | 52.71 | 48.86 | 37.86 | 26.17 | 26.60 | 20.39 | 13.0 | 32.23 |
| | w. OWL | 52.71 | 49.49 | 37.83 | 26.62 | 26.94 | 19.11 | 12.6 | 32.19 |
| | AlphaPruning | 52.71 | 50.67 | 38.01 | 26.62 | 26.77 | 20.14 | 11.2 | 32.30 |
| | w. NEURONAL | 52.71 | 49.17 | 38.17 | 28.1 | 29.46 | 18.34 | 15.0 | **32.99** |
| | Wanda | 47.65 | 49.41 | 37.83 | 25.82 | 26.52 | **20.82** | 14.6 | 31.81 |
| | w. DSnoT | **53.07** | 47.91 | 37.86 | 26.09 | 27.23 | 20.73 | 13.0 | 32.27 |
| | w. OWL | 52.71 | **50.83** | 37.83 | 26.52 | 27.27 | 19.37 | 12.8 | 32.48 |
| | AlphaPruning | 52.71 | 48.54 | 37.89 | 26.42 | 27.44 | 19.45 | 12.2 | 32.09 |
| | w. NEURONAL | 52.71 | 50.04 | 37.77 | 27.3 | 28.45 | 18.94 | 14.0 | **32.74** |
| Mistral-7B | MULTIFLOW | 50.18 | 48.15 | 37.80 | 25.67 | 26.18 | 22.70 | 13.20 | 31.98 |
| | w. DSnoT | **52.71** | 47.36 | 37.83 | **26.58** | **28.03** | 18.94 | **13.8** | 32.18 |
| | w. OWL | 48.38 | 49.09 | **38.44** | 25.88 | 25.59 | **23.29** | 13.6 | 32.04 |
| | AlphaPruning | 51.62 | 50.67 | 37.92 | 26.35 | 26.35 | 21.67 | 14.8 | **32.77** |
| | w. NEURONAL | 52.71 | 48.46 | 37.83 | 26.07 | 27.57 | 19.03 | 14.6 | 32.32 |
| | Wanda | **53.79** | 48.78 | 37.83 | 26.52 | 27.82 | 19.8 | 12.8 | 32.48 |
| | w. DSnoT | 52.35 | 48.30 | 37.83 | 26.55 | 27.44 | 19.54 | 13.0 | 32.14 |
| | w. OWL | 52.71 | 47.43 | 37.83 | 26.68 | 27.78 | 18.52 | 13.2 | 32.02 |
| | AlphaPruning | 52.71 | 47.99 | 37.83 | 26.73 | 28.41 | 19.54 | 12.8 | 32.29 |
| | w. NEURONAL | 52.71 | 51.14 | 38.04 | 27.28 | 28.66 | 20.31 | 13.4 | **33.08** |
| OPT-6.7B | MULTIFLOW | 52.71 | 50.91 | 37.80 | 25.87 | 27.40 | 19.28 | 12.40 | 32.34 |
| | w. DSnoT | 52.71 | 50.83 | **57.31** | 26.00 | 25.00 | **20.22** | 11.80 | 34.84 |
| | w. OWL | 52.71 | 51.07 | 37.83 | 25.74 | 25.29 | 20.05 | **16.00** | 32.67 |
| | AlphaPruning | 52.71 | 49.01 | 37.83 | 26.14 | 27.31 | 20.73 | 14.8 | 32.65 |
| | w. NEURONAL | 53.07 | 50.12 | 62.23 | 26.44 | 31.48 | 20.48 | 12.6 | **36.63** |
| | Wanda | **54.15** | **52.09** | 41.53 | **26.47** | 28.45 | 18.86 | 11.80 | 33.34 |
| | w. DSnoT | 52.71 | 51.38 | **55.32** | 26.17 | 27.06 | 19.37 | 13.80 | **35.12** |
| | w. OWL | 52.71 | 49.33 | 37.83 | 25.84 | 25.67 | **20.31** | **13.00** | 32.10 |
| | AlphaPruning | 52.71 | 48.86 | 37.98 | 26.09 | 26.73 | 20.14 | 12.6 | 32.16 |
| | w. NEURONAL | 52.71 | 50.67 | 41.83 | 26.43 | 30.05 | 18.69 | 11.0 | 33.05 |

### D.4 Zero-Shot at 60%, 70%, and 80% Sparsity with Magnitude

In Tables 22-24, we report the results of NEURONAL over Zero-Shot tasks using Magnitude pruning.

The results provided by NEURONAL turn out to be the best in 10 out of 15 cases, while being the second best in 3 cases. It is also worth noticing that the performance gap between the Magnitude pruning and score-based pruning algorithms (such as Wanda or MULTIFLOW) is generally quite high. Hence, NEURONAL can improve the performance of Magnitude (in the standard setting with uniform distribution) only to a certain degree, since at high sparsity ratios (as the ones we test) the performance of Magnitude has been shown to be poor (Jaiswal et al., 2024).

Table 22: Accuracy on the seven Zero-Shot Tasks, computed over five different LLMs for three different top-up pruning algorithms (DSnoT, OWL, and NEURONAL) on Magnitude at 60% sparsity. "Average" indicates the mean accuracy across tasks. The rows corresponding to the pruning algorithms refer to the uniform distribution.

| Model | Algorithm | RTE | WinoGrande | BoolQ | HellaSwag | ARC-e | ARC-c | OBQA | Average |
|---|---|---|---|---|---|---|---|---|---|
| Phi-2.7B | Magnitude | **57.04** | 62.83 | **51.38** | **42.56** | **66.33** | 35.41 | 28.2 | **49.11** |
| | w. DSnoT | 54.51 | 64.09 | 42.32 | 41.09 | 66.25 | 34.98 | 26.6 | 47.12 |
| | w. OWL | 55.23 | 62.59 | 48.81 | 42.53 | 67.38 | **38.48** | **28.4** | 49.06 |
| | w. AlphaPruning | 57.76 | 56.2 | 47.34 | 35.16 | 58.46 | 33.28 | 23.2 | 44.49 |
| | w. NEURONAL | 54.15 | 65.67 | 47.43 | 42.38 | 65.95 | 36.77 | 26.8 | 48.45 |
| LLama-1 7B | Magnitude | 51.62 | 52.64 | 45.05 | 39.23 | 51.05 | 26.88 | 20.4 | 40.98 |
| | w. DSnoT | **52.35** | 52.80 | 46.88 | 38.3 | 50.59 | 26.37 | 20.6 | 41.13 |
| | w. OWL | **52.35** | **58.41** | 51.8 | **42.02** | **56.31** | **29.78** | 23.8 | 44.92 |
| | w. AlphaPruning | 53.79 | 57.14 | 56.36 | 40.83 | 56.82 | 32.25 | 24.4 | **45.94** |
| | w. NEURONAL | 50.54 | 56.04 | 57.46 | 40.63 | 55.26 | 29.86 | 24.4 | 44.88 |
| LLama-2 7B | Magnitude | 51.26 | 55.8 | 41.19 | 36.97 | 50.17 | 26.96 | 16.2 | 39.79 |
| | w. DSnoT | 53.79 | 56.04 | 42.87 | 38.3 | 53.28 | 27.9 | 19.8 | 41.71 |
| | w. OWL | 51.99 | 57.3 | 46.15 | 42.56 | 56.65 | 30.46 | 19.4 | 43.50 |
| | w. AlphaPruning | 52.35 | 61.88 | 58.1 | 46.13 | 58.59 | 31.83 | 24.8 | 47.67 |
| | w. NEURONAL | 55.23 | 59.59 | 60.43 | 46.15 | 58.96 | 32.85 | 27.6 | **48.69** |
| Mistral-7B | Magnitude | **55.23** | 62.19 | 66.36 | 48.74 | 67.05 | 33.19 | 22.6 | 50.77 |
| | w. DSnoT | 55.6 | 62.35 | 68.53 | 48.28 | 67.51 | 33.11 | 23.2 | 51.23 |
| | w. OWL | 53.79 | **64.48** | **72.17** | 49.39 | **68.14** | 33.87 | 23.8 | 52.23 |
| | w. AlphaPruning | 54.87 | 63.93 | 74.89 | 47.24 | 63.89 | 32.0 | 23.0 | 51.40 |
| | w. NEURONAL | 54.15 | 64.96 | 71.62 | 49.82 | 65.45 | 35.92 | 24.6 | **52.36** |
| OPT-6.7B | Magnitude | **53.43** | 50.59 | 37.86 | 26.38 | 26.6 | 21.42 | 13.2 | 32.78 |
| | w. DSnoT | 52.71 | 49.25 | 37.86 | 26.14 | 27.27 | 21.5 | 13.2 | 32.56 |
| | w. OWL | 52.71 | 50.51 | 37.83 | 26.77 | 30.3 | 18.52 | 14.8 | 33.06 |
| | w. AlphaPruning | 52.71 | 51.22 | 37.83 | 26.54 | 29.84 | 19.8 | 13.4 | 33.05 |
| | w. NEURONAL | 52.71 | 53.91 | 39.11 | 33.23 | 37.71 | 24.06 | 16.8 | **36.79** |

Table 23: Accuracy on the seven Zero-Shot Tasks, computed over five different LLMs for three different top-up pruning algorithms (DSnoT, OWL, and NEURONAL) on Magnitude at 70% sparsity. "Average" indicates the mean accuracy across tasks. The rows corresponding to the pruning algorithms refer to the uniform distribution.

| Model | Algorithm | RTE | WinoGrande | BoolQ | HellaSwag | ARC-e | ARC-c | OBQA | Average |
|---|---|---|---|---|---|---|---|---|---|
| Phi-2.7B | Magnitude | 46.93 | 53.59 | 47.22 | 30.45 | 47.85 | 24.57 | 19.2 | 38.54 |
| | w. DSnoT | 46.57 | 50.91 | 39.6 | 30.12 | 45.54 | 24.06 | 16.8 | 36.23 |
| | w. OWL | 45.13 | 52.88 | 49.2 | 32.26 | 51.64 | 27.56 | 21.4 | 40.01 |
| | w. AlphaPruning | 47.65 | 52.41 | 38.01 | 26.06 | 25.25 | 22.53 | 13.2 | 32.16 |
| | w. NEURONAL | 47.65 | 53.51 | 52.81 | 33.22 | 53.87 | 30.03 | 20.6 | **41.67** |
| LLama-1 7B | Magnitude | **53.43** | 49.96 | 37.92 | 27.59 | 31.73 | 22.44 | 16.6 | 34.24 |
| | w. DSnoT | 52.71 | 51.7 | 37.83 | 27.71 | 30.26 | 22.7 | 15.4 | 34.04 |
| | w. OWL | 53.07 | 51.38 | 38.38 | 33.14 | 39.31 | 24.15 | 16.8 | 36.6 |
| | w. AlphaPruning | 52.71 | 53.04 | 39.48 | 36.18 | 43.56 | 24.83 | 21.2 | 38.71 |
| | w. NEURONAL | 52.71 | 54.62 | 52.6 | 39.81 | 46.13 | 26.79 | 23.0 | **42.24** |
| LLama-2 7B | Magnitude | 51.26 | 49.96 | 37.86 | 25.9 | 28.45 | 23.12 | 13.4 | 32.85 |
| | w. DSnoT | 53.79 | 49.88 | 37.86 | 25.42 | 28.83 | 20.56 | 16.6 | 33.28 |
| | w. OWL | 53.07 | 50.28 | 37.89 | 26.38 | 30.77 | 22.7 | 15.0 | 33.73 |
| | w. AlphaPruning | 52.71 | 50.83 | 43.91 | 35.01 | 41.67 | 25.34 | 19.8 | 38.47 |
| | w. NEURONAL | 54.51 | 55.41 | 64.86 | 33.25 | 42.09 | 27.47 | 21.2 | **42.68** |
| Mistral-7B | Magnitude | 51.99 | 50.83 | 41.13 | 32.16 | 42.72 | 19.54 | 16.6 | 36.42 |
| | w. DSnoT | 53.07 | 51.62 | 39.54 | 31.66 | 42.51 | 20.05 | 16.6 | 36.44 |
| | w. OWL | **57.76** | 56.59 | 49.17 | 36.48 | **45.75** | 22.01 | 18.8 | 40.94 |
| | w. AlphaPruning | 53.07 | 58.96 | 57.71 | 34.47 | 42.3 | 22.44 | 16.0 | 40.71 |
| | w. NEURONAL | 53.79 | 58.56 | 62.6 | 38.6 | 44.23 | 26.28 | 21.0 | **43.58** |
| OPT-6.7B | Magnitude | 52.71 | 49.8 | 37.83 | 25.88 | 26.68 | **21.33** | 12.4 | 32.38 |
| | w. DSnoT | 52.71 | 49.96 | 37.83 | 25.87 | **27.19** | 20.14 | **13.6** | 32.47 |
| | w. OWL | 52.71 | **50.59** | 37.83 | 25.81 | 25.46 | 21.25 | 12.8 | 32.35 |
| | w. AlphaPruning | 52.71 | 52.01 | 37.83 | 26.21 | 27.74 | 20.9 | 13.0 | **32.91** |
| | w. NEURONAL | 52.71 | 50.43 | 37.83 | 26.25 | 26.89 | 20.39 | 13.0 | 32.50 |

Table 24: Accuracy on the seven Zero-Shot Tasks, computed over five different LLMs for three different top-up pruning algorithms (DSnoT, OWL, and NEURONAL) on Magnitude at 80% sparsity. "Average" indicates the mean accuracy across tasks. The rows corresponding to the pruning algorithms refer to the uniform distribution.

| Model | Algorithm | RTE | WinoGrande | BoolQ | HellaSwag | ARC-e | ARC-c | OBQA | Average |
|---|---|---|---|---|---|---|---|---|---|
| Phi-2.7B | Magnitude | 45.13 | 50.36 | 41.19 | 25.83 | 29.08 | 20.9 | **13.6** | 32.30 |
| | w. DSnoT | 46.93 | **52.33** | 39.63 | 25.9 | 28.32 | 21.25 | 13.4 | 32.54 |
| | w. OWL | **49.46** | 50.91 | 42.35 | **26.71** | **35.27** | **21.67** | 13.4 | **34.25** |
| | w. AlphaPruning | 49.82 | 49.25 | 47.28 | 25.85 | 25.38 | 21.84 | 16.6 | 33.72 |
| | w. NEURONAL | 50.54 | 52.25 | 42.69 | 26.21 | 28.7 | 22.1 | 12.8 | 33.61 |
| LLama-1 7B | Magnitude | 46.21 | 49.96 | **53.98** | 25.69 | 24.83 | **21.84** | 13.8 | **33.76** |
| | w. DSnoT | **52.35** | 51.85 | 38.47 | 25.52 | 26.39 | 21.42 | 16.0 | 33.14 |
| | w. OWL | 48.38 | 48.93 | 44.74 | **25.76** | 26.35 | 21.08 | **15.8** | 33.01 |
| | w. AlphaPruning | 51.26 | 50.2 | 39.54 | 26.25 | 28.28 | 20.99 | 15.8 | 33.19 |
| | w. NEURONAL | 47.29 | 48.78 | 50.31 | 25.8 | 26.09 | 21.25 | 14.6 | 33.45 |
| LLama-2 7B | Magnitude | 52.35 | 49.57 | 46.18 | **25.94** | 26.14 | **23.12** | **16.0** | 34.19 |
| | w. DSnoT | 52.71 | **51.54** | 37.89 | 25.46 | **27.10** | 22.44 | 15.4 | 33.22 |
| | w. OWL | **53.07** | 48.70 | 42.02 | 25.72 | 26.60 | 21.42 | 14.4 | 33.13 |
| | w. AlphaPruning | 55.23 | 49.33 | 40.37 | 25.84 | 26.56 | 22.01 | 16.2 | 33.65 |
| | w. NEURONAL | 54.15 | 50.36 | 52.69 | 26.82 | 29.67 | 20.14 | 14.6 | **35.49** |
| Mistral-7B | Magnitude | 51.26 | **50.99** | 41.16 | 25.93 | 27.48 | **21.84** | 14.6 | 33.32 |
| | w. DSnoT | 52.35 | 49.72 | 38.07 | 26.26 | 26.43 | 21.25 | 14.0 | 32.58 |
| | w. OWL | 52.35 | 50.20 | 41.04 | 26.55 | 27.78 | 19.97 | 13.8 | 33.10 |
| | w. AlphaPruning | 53.07 | 47.67 | 37.86 | 26.4 | 28.45 | 19.54 | 13.4 | 32.34 |
| | w. NEURONAL | 53.43 | 50.04 | 61.04 | 28.56 | 32.2 | 22.53 | 16.2 | **37.71** |
| OPT-6.7B | Magnitude | 52.71 | 49.49 | 37.83 | 25.79 | 26.39 | **21.25** | 13.0 | 32.35 |
| | w. DSnoT | 52.71 | 49.57 | 37.83 | 25.78 | 25.63 | 20.65 | 12.8 | 32.14 |
| | w. OWL | 52.71 | 49.80 | 37.83 | **26.05** | 26.73 | 21.16 | **13.2** | **32.50** |
| | w. AlphaPruning | 52.71 | 49.09 | 37.83 | 25.84 | 26.14 | 20.82 | 12.8 | 32.18 |
| | w. NEURONAL | 52.71 | 50.91 | 37.83 | 25.78 | 26.73 | 20.9 | 12.4 | 32.47 |

**D.5   Zero-Shot at 70% and 80% Sparsity for LLama 13B (both v1 and v2) and 70B (v2)**

In Tables 25-27, we report the single-task detailed results for the Zero-Shot tasks, for which only the averaged results are reported in the main text in Table 9. These detailed results clearly show the gap between NEURONAL and the baselines: as mentioned in the main text, NEURONAL turns out to be the *top-up* algorithm that, in the majority of the cases, provides the strongest results across all tested models.

Table 25: Accuracy on the seven Zero-Shot Tasks, computed over LLama-1 13B for four different top-up pruning algorithms (DSnoT, OWL, AlphaPruning, and NEURONAL) on two pruning algorithms (MULTIFLOW and Wanda) at 70% and 80% sparsity. "Average" indicates the mean accuracy across tasks. The rows corresponding to the pruning algorithms refer to the uniform distribution.

| Sparsity | Algorithm | RTE | WinoGrande | BoolQ | HellaSwag | ARC-e | ARC-c | OBQA | Average |
|---|---|---|---|---|---|---|---|---|---|
| Dense | | 71.48 | 72.53 | 78.04 | 59.84 | 77.31 | 46.25 | 33.0 | 62.64 |
| 70% | MULTIFLOW | 52.35 | 54.30 | 61.93 | 32.22 | 45.41 | 19.97 | 16.80 | 40.43 |
| | w. DSnoT | 52.71 | 53.04 | 62.17 | 31.38 | 43.01 | 19.97 | 17.20 | 39.93 |
| | w. OWL | 52.71 | 63.93 | 67.28 | 40.63 | 57.83 | 27.99 | 20.80 | 47.31 |
| | AlphaPruning | 54.15 | **66.06** | 66.30 | 43.04 | 57.70 | 28.92 | 21.40 | 48.22 |
| | w. NEURONAL | 54.15 | 65.90 | **69.72** | **45.10** | **62.54** | **30.89** | **27.20** | **50.79** |
| | Wanda | 52.71 | 53.43 | 61.74 | 30.61 | 40.87 | 16.72 | 16.60 | 38.95 |
| | w. DSnoT | 52.71 | 53.12 | 62.20 | 30.41 | 39.98 | 17.75 | 16.00 | 38.88 |
| | w. OWL | 52.71 | 63.85 | 63.55 | 39.75 | 57.74 | 27.22 | 20.80 | 46.52 |
| | AlphaPruning | 53.43 | 65.75 | 62.78 | 42.23 | 58.12 | 28.33 | 21.20 | 47.41 |
| | w. NEURONAL | 52.71 | **64.64** | **65.75** | **43.96** | **61.49** | **30.80** | **26.00** | **49.34** |
| 80% | MULTIFLOW | **53.79** | 49.01 | 37.83 | 26.89 | 26.98 | 20.99 | 14.00 | 32.78 |
| | w. DSnoT | 52.71 | 49.64 | 38.17 | 26.99 | 27.31 | 19.80 | 12.40 | 32.43 |
| | w. OWL | 52.71 | 49.49 | 38.59 | 27.64 | 28.45 | 18.17 | 10.80 | 32.26 |
| | AlphaPruning | 52.71 | 53.04 | **62.17** | 30.46 | 33.75 | 22.18 | 15.60 | 38.56 |
| | w. NEURONAL | 52.71 | **55.09** | **62.17** | **32.21** | **39.48** | **23.81** | **19.60** | **40.72** |
| | Wanda | 52.71 | 50.67 | 37.83 | 26.68 | 27.23 | 19.97 | 14.20 | 32.76 |
| | w. DSnoT | 52.71 | 50.04 | 37.83 | 26.83 | 27.90 | 18.94 | 12.60 | 32.41 |
| | w. OWL | 52.71 | 48.62 | 37.89 | 27.37 | 28.45 | 18.52 | 12.20 | 32.25 |
| | AlphaPruning | 52.71 | **55.25** | **62.17** | **29.56** | 34.30 | **20.31** | 12.20 | **38.07** |
| | w. NEURONAL | 52.71 | 51.14 | 62.11 | 29.42 | **34.72** | 19.97 | 15.80 | 37.98 |

Table 26: Accuracy on the seven Zero-Shot Tasks, computed over LLama-2 13B for four different top-up pruning algorithms (DSnoT, OWL, AlphaPruning, and NEURONAL) on two pruning algorithms (MULTIFLOW and Wanda) at 70% and 80% sparsity. "Average" indicates the mean accuracy across tasks. The rows corresponding to the pruning algorithms refer to the uniform distribution.

| Sparsity | Algorithm | RTE | WinoGrande | BoolQ | HellaSwag | ARC-e | ARC-c | OBQA | Average |
|---|---|---|---|---|---|---|---|---|---|
| Dense | | 65.34 | 72.14 | 80.61 | 60.06 | 79.38 | 48.46 | 35.2 | 63.03 |
| 70% | MULTIFLOW | 52.71 | 49.25 | 39.82 | 27.34 | 29.21 | 19.20 | 11.60 | 32.73 |
| | w. DSnoT | 52.71 | 51.22 | 62.05 | 28.75 | 34.68 | 17.49 | 11.80 | 36.96 |
| | w. OWL | 52.71 | 51.85 | 60.55 | 28.60 | 39.44 | 19.20 | 14.00 | 38.05 |
| | AlphaPruning | 52.71 | **63.85** | 62.45 | 34.79 | 53.58 | 26.45 | 20.80 | 44.95 |
| | w. NEURONAL | 52.71 | 62.51 | **70.06** | **38.66** | **57.70** | **28.07** | **25.00** | **47.82** |
| | Wanda | 52.71 | 51.78 | 62.23 | 29.09 | 36.70 | 17.66 | 12.60 | 37.54 |
| | w. DSnoT | 52.71 | 50.67 | 62.11 | 28.78 | 36.62 | 17.66 | 12.80 | 37.34 |
| | w. OWL | 52.71 | 61.25 | 65.14 | 38.45 | 57.70 | 26.88 | 22.20 | 46.33 |
| | AlphaPruning | 52.71 | **67.48** | 63.64 | 40.21 | 56.73 | 29.27 | 22.20 | 47.46 |
| | w. NEURONAL | 52.71 | 62.04 | **70.67** | **40.28** | **58.59** | **29.69** | **24.00** | **48.28** |
| 80% | MULTIFLOW | 52.71 | 49.96 | 37.83 | 26.16 | 26.01 | 20.90 | 13.20 | 32.4 |
| | w. DSnoT | 52.71 | 50.20 | 38.26 | 26.09 | 28.41 | 20.65 | 11.80 | 32.59 |
| | w. OWL | 52.71 | 50.28 | 37.83 | 26.41 | 27.36 | 20.90 | 11.00 | 32.36 |
| | AlphaPruning | 52.71 | **49.64** | **37.95** | 26.63 | 27.10 | 20.31 | 11.60 | 32.28 |
| | w. NEURONAL | 52.71 | 48.78 | 37.92 | **26.45** | **27.48** | **20.82** | **13.40** | **32.51** |
| | Wanda | 52.35 | 52.57 | 37.83 | 26.52 | 26.64 | 20.05 | 13.80 | 32.82 |
| | w. DSnoT | 52.71 | 49.25 | 37.83 | 26.02 | 26.64 | 20.14 | 12.40 | 32.14 |
| | w. OWL | 52.71 | 48.30 | 41.35 | 27.09 | 27.48 | 19.11 | 11.40 | 32.49 |
| | AlphaPruning | 52.71 | **52.64** | **62.23** | 27.74 | **30.13** | **19.71** | 11.00 | **36.59** |
| | w. NEURONAL | 52.35 | 50.99 | 61.65 | **28.16** | 29.59 | 18.17 | 12.40 | 36.19 |

Table 27: Accuracy on the seven Zero-Shot Tasks, computed over LLama-2 70B for four different top-up pruning algorithms (DSnoT, OWL, AlphaPruning, and NEURONAL) on two pruning algorithms (MULTIFLOW and Wanda) at 70% and 80% sparsity. "Average" indicates the mean accuracy across tasks. The rows corresponding to the pruning algorithms refer to the uniform distribution.

| Sparsity | Algorithm | RTE | WinoGrande | BoolQ | HellaSwag | ARC-e | ARC-c | OBQA | Average |
|---|---|---|---|---|---|---|---|---|---|
| Dense | | 67.87 | 77.98 | 83.5 | 66.10 | 82.60 | 54.44 | 37.20 | 67.10 |
| 70% | MULTIFLOW | 56.32 | 71.51 | 73.95 | 49.00 | 70.30 | 37.29 | 26.80 | 55.02 |
| | w. DSnoT | 59.21 | **74.43** | 75.35 | 51.50 | 72.45 | **39.76** | **28.8** | **57.36** |
| | w. OWL | 61.01 | 74.11 | 72.40 | 51.10 | **72.70** | 37.71 | 27.20 | 56.60 |
| | AlphaPruning | **61.37** | 73.72 | 74.50 | 50.25 | 71.75 | 38.05 | 28.20 | 56.83 |
| | w. NEURONAL | 61.01 | 72.77 | **75.90** | **52.20** | 71.25 | 37.29 | 28.20 | 56.95 |
| | Wanda | 61.01 | 74.43 | 74.65 | 48.80 | 71.75 | **39.25** | 27.20 | 56.73 |
| | w. DSnoT | 57.40 | 72.85 | 75.65 | 49.25 | 71.85 | 38.82 | 26.80 | 56.09 |
| | w. OWL | **64.98** | 75.22 | 72.90 | 50.95 | 73.40 | 38.14 | **29.80** | 57.91 |
| | AlphaPruning | 63.18 | 74.98 | 73.55 | 51.30 | 72.65 | 38.31 | 29.40 | 57.62 |
| | w. NEURONAL | 62.09 | **75.93** | **78.90** | **52.80** | **74.40** | 38.99 | 29.60 | **58.96** |
| 80% | MULTIFLOW | 52.35 | 48.62 | 62.60 | 27.35 | 26.00 | 18.17 | 11.20 | 35.18 |
| | w. DSnoT | 47.65 | 49.01 | 62.60 | 27.70 | 27.80 | 17.15 | 11.80 | 34.82 |
| | w. OWL | 46.93 | 55.41 | 62.75 | 31.10 | 37.90 | 19.80 | 13.40 | 38.18 |
| | AlphaPruning | 46.93 | 60.93 | 62.65 | 33.30 | **47.85** | 23.38 | 16.40 | 41.63 |
| | w. NEURONAL | **54.51** | **65.04** | **65.55** | **38.40** | 46.55 | **27.99** | **19.20** | **45.32** |
| | Wanda | 50.18 | 49.49 | 62.30 | 26.95 | 26.80 | 17.92 | 11.20 | 34.98 |
| | w. DSnoT | 46.93 | 49.25 | 61.70 | 27.15 | 27.40 | 18.09 | 12.00 | 34.65 |
| | w. OWL | 49.82 | 57.38 | 62.70 | 31.30 | 40.10 | 20.14 | 13.00 | 39.21 |
| | AlphaPruning | 48.01 | 61.96 | 62.65 | 33.80 | 48.75 | 22.44 | 16.40 | 42.00 |
| | w. NEURONAL | **55.96** | **68.43** | **64.90** | **39.55** | **49.45** | **29.44** | **19.80** | **46.79** |

### D.6 NeuronAL on SparseGPT at 60%, 70%, and 80% Sparsity

In Tables 28-30, we present the results of NEURONAL on SparseGPT on the WikiText2, C4, and PTB datasets, using the *block*-only setting. To note: Since SparseGPT prunes and updates the weights from columns to rows, the *row* step of NEURONAL cannot be included: indeed, it would force each row to have a different sparsity ratio, which is in contrast with the nature of SparseGPT.

Using SparseGPT, the superiority of NEURONAL is less evident than with other pruning algorithms. Nevertheless, NEURONAL turns out to be the best-performing top-up algorithm in 5 out of 15, 8 out of 15, and 7 out of 15 cases, respectively, for WikiText2, C4, and PTB. Interestingly, for lower sparsity, the gap between uniform and non-uniform distribution (both NEURONAL and OWL) is less remarkable than at higher sparsity. We explain these results with the inherent functioning of SparseGPT, which, differently from the other pruning algorithms, includes a weight reconstruction step. However, we can conclude that, also in this case, our proposed approach turns out to be effective in many cases at increasing the task performance.

Table 28: Perplexity on WikiText2 using SparseGPT.

| Sparsity | Top-Up | Model | | | | |
|---|---|---|---|---|---|---|
| | | Phi-2.7B | LLama-1 7B | LLama-2 7B | Mistral-7B | OPT-6.7B |
| 60% | Uniform | 15.8 | 10.4 | 10.2 | 9.4 | **13.4** |
| | OWL | 15.8 | **9.4** | **9.2** | 9.1 | 14.2 |
| | NEURONAL | **15.7** | 9.9 | 9.3 | **9.1** | 13.7 |
| 70% | Uniform | 28.9 | 27.3 | 27.3 | 22.0 | **20.5** |
| | OWL | 27.7 | **19.2** | **20.5** | 18.6 | 21.6 |
| | NEURONAL | **27.3** | 22.6 | 20.9 | **17.8** | 21.8 |
| 80% | Uniform | 131.0 | 207.0 | 122.1 | 98.4 | 95.7 |
| | OWL | **107.5** | **93.8** | **84.3** | 77.2 | **80.8** |
| | NEURONAL | 113.5 | 144.7 | 88.7 | **70.8** | 84.0 |

Table 29: Perplexity on C4 using SparseGPT.

| Sparsity | Top-Up | Model | | | | |
|---|---|---|---|---|---|---|
| | | Phi-2.7B | LLama-1 7B | LLama-2 7B | Mistral-7B | OPT-6.7B |
| 60% | Uniform | **19.0** | 12.8 | 12.9 | 13.0 | **15.3** |
| | OWL | 19.2 | **11.7** | **11.6** | 12.4 | 15.8 |
| | NEURONAL | 19.1 | 12.4 | 11.7 | **12.3** | 15.5 |
| 70% | Uniform | 28.6 | 28.3 | 31.5 | 27.8 | 22.4 |
| | OWL | 28.2 | **21.1** | 22.8 | 23.7 | 22.4 |
| | NEURONAL | **27.8** | 23.8 | **22.5** | **21.9** | **22.2** |
| 80% | Uniform | 98.7 | 136.2 | 104.8 | 86.5 | 72.5 |
| | OWL | **79.7** | **68.3** | 73.4 | 66.2 | 65.4 |
| | NEURONAL | 86.4 | 104.2 | **72.4** | **61.8** | **65.0** |

Table 30: Perplexity on PTB using SparseGPT.

| Sparsity | Top-Up | Model | | | | |
|---|---|---|---|---|---|---|
| | | Phi-2.7B | LLama-1 7B | LLama-2 7B | Mistral-7B | OPT-6.7B |
| 60% | Uniform | 28.7 | 19.5 | 430.5 | 73.7 | **20.3** |
| | OWL | 29.3 | **16.9** | 262.1 | 70.9 | 21.0 |
| | NEURONAL | **28.2** | 18.2 | **249.2** | **67.2** | 20.6 |
| 70% | Uniform | 50.3 | 52.6 | 3780.0 | 153.2 | 32.0 |
| | OWL | 51.0 | **37.0** | 1643.4 | 135.0 | **32.9** |
| | NEURONAL | **47.3** | 40.5 | **861.6** | **123.4** | 32.8 |
| 80% | Uniform | 195.4 | 295.6 | **3201.7** | 316.2 | 102.3 |
| | OWL | **141.4** | **162.3** | 5556.5 | 278.8 | **98.9** |
| | NEURONAL | 156.7 | 260.2 | 3659.8 | **266.6** | 105.3 |

## D.7 Results on Broader Sparsity Range

While in the main text, we mainly tested high sparsity configurations (hence $s \geq 0.6$), here we report the results of NEURONAL as well as the baselines for several sparsity values between 10% and 80%. Table 31, for LLama-2 7B, and Table 32, for Phi-2, show the results of perplexity across three datasets (Wikitext, C4, and PTB) for different sparsity ratios. The algorithm's setup has been kept the same as the experiments in the main text, see Appendix C. The results provide a non-trivial pattern: for up to 50% sparsity, all the *top-up* algorithms perform similarly (with very small deviations), with DSnoT providing almost in all cases the best results. At such low sparsity values, even Uniform achieves effective results.

However, the pattern completely changes for sparsity above 60%, where, as already shown in the main text, NEURONAL turns out to be the most reliable among all the tested *top-up* algorithms in achieving the best performance. This trend is in line with the results provided in the original baseline papers (Yin et al., 2024; Lu et al., 2024), where the authors also recognize this sparsity-performance pattern for non-uniform sparsity *top-up* algorithms.

Table 31: Perplexity di LLama-2 7B at different sparsity levels (10%–80%) on the WikiText2, C4, and PTB datasets.

| Top-Up | WikiText2 | | | | | | | |
|---|---|---|---|---|---|---|---|---|
| | 10% | 20% | 30% | 40% | 50% | 60% | 70% | 80% |
| Uniform | 5.49 | 5.59 | 5.74 | 6.06 | 6.92 | 10.77 | 78.01 | 4.93e3 |
| OWL | 5.49 | 5.59 | 5.76 | 6.10 | 6.86 | **9.17** | 24.9 | 663.0 |
| DSnoT | **5.48** | **5.52** | **5.65** | **5.96** | **6.85** | 10.79 | 76.11 | 5.20e3 |
| AlphaPruning | 5.49 | 5.59 | 5.77 | 6.14 | 7.06 | 9.82 | 31.97 | 982.08 |
| **NeuronAL** | 5.50 | 5.58 | 5.74 | 6.09 | 6.92 | 9.38 | **24.05** | **557.16** |

| Top-Up | C4 | | | | | | | |
|---|---|---|---|---|---|---|---|---|
| | 10% | 20% | 30% | 40% | 50% | 60% | 70% | 80% |
| Uniform | 7.28 | 7.41 | 7.63 | 8.10 | 9.24 | 13.99 | 81.01 | 3.12e3 |
| OWL | 7.29 | 7.43 | 7.66 | 8.17 | 9.22 | **11.85** | 30.49 | 662.95 |
| DSnoT | **7.27** | **7.33** | **7.50** | **7.96** | **9.12** | 14.11 | 85.65 | 4.44e3 |
| AlphaPruning | 7.29 | 7.42 | 7.68 | 8.27 | 9.50 | 12.62 | 37.73 | 670.03 |
| **NeuronAL** | 7.31 | 7.40 | 7.63 | 8.15 | 9.26 | 11.99 | **27.42** | **660.21** |

| Top-Up | PTB | | | | | | | |
|---|---|---|---|---|---|---|---|---|
| | 10% | 20% | 30% | 40% | 50% | 60% | 70% | 80% |
| Uniform | 33.07 | 33.65 | 34.55 | 38.02 | 48.20 | 122.25 | 599.27 | 5.29e3 |
| OWL | 33.01 | 33.45 | 34.54 | 36.65 | **43.16** | 75.09 | 333.7 | 2.28e3 |
| DSnoT | **32.81** | **32.98** | **34.10** | 37.25 | 47.08 | 109.61 | 491.77 | 6.69e3 |
| AlphaPruning | 33.07 | 33.83 | 34.72 | 37.20 | 43.85 | 69.34 | 273.84 | **2.18e3** |
| **NeuronAL** | 33.15 | 33.74 | 34.60 | **36.60** | 43.33 | **64.34** | **206.97** | 2.40e3 |

Table 32: Perplexity of Phi-2 at different sparsity levels (10 %–80 %) on the WikiText2, C4, and PTB datasets.

| Top-Up | WikiText2 | | | | | | | |
|---|---|---|---|---|---|---|---|---|
| | 10% | 20% | 30% | 40% | 50% | 60% | 70% | 80% |
| Uniform | 9.85 | 10.07 | 10.50 | **11.46** | 14.22 | 25.78 | 227.56 | 2.04e4 |
| OWL | **9.83** | **10.05** | 10.48 | 11.47 | 14.20 | **24.80** | 132.65 | 2.54e3 |
| DSnoT | 9.88 | 10.19 | **10.75** | 12.02 | 15.57 | 32.21 | 221.86 | 1.52e4 |
| AlphaPruning | 9.87 | 10.09 | 10.59 | 12.05 | 16.66 | 165.29 | 4.22e4 | 4.30e4 |
| **NeuronAL** | 9.86 | 10.08 | 10.51 | 11.59 | 14.52 | 25.26 | **88.32** | **2.49e3** |

| Top-Up | C4 | | | | | | | |
|---|---|---|---|---|---|---|---|---|
| | 10% | 20% | 30% | 40% | 50% | 60% | 70% | 80% |
| Uniform | 14.25 | 14.48 | **14.87** | 15.81 | 18.25 | 29.28 | 182.70 | 1.24e4 |
| OWL | **14.22** | 14.46 | **14.87** | 15.78 | 18.24 | **28.18** | 116.21 | **1.21e3** |
| DSnoT | 14.29 | 14.74 | 15.20 | 16.54 | 20.22 | 38.02 | 172.61 | 6.86e3 |
| AlphaPruning | 14.25 | 14.50 | 14.98 | 16.31 | 20.26 | 166.48 | 3.05e4 | 3.33e4 |
| **NeuronAL** | 14.26 | 14.49 | 14.89 | 15.89 | 18.54 | **27.10** | **77.69** | 1.59e3 |

| Top-Up | PTB | | | | | | | |
|---|---|---|---|---|---|---|---|---|
| | 10% | 20% | 30% | 40% | 50% | 60% | 70% | 80% |
| Uniform | 18.37 | **18.70** | **19.47** | 21.13 | 25.67 | 48.89 | 346.16 | 3.14e4 |
| OWL | **18.34** | **18.70** | 19.48 | **21.02** | 25.68 | 48.63 | 183.67 | 7.06e3 |
| DSnoT | 18.45 | 18.85 | 19.75 | 21.67 | 27.26 | 50.58 | 257.60 | 1.40e4 |
| AlphaPruning | 18.35 | 18.74 | 19.65 | 21.89 | 28.69 | 669.69 | 2.24e4 | 2.66e4 |
| **NeuronAL** | 18.37 | 18.71 | 19.49 | 21.22 | **25.62** | **41.78** | **129.53** | **4.03e3** |

## D.8 Sensitivity to Calibration Data

In Tables 33-34, we complement the results regarding the seed set of the calibration data at 60% and 80% sparsity. The results are fully in line with the ones presented in the main text. As expected, the standard deviation of the performance increases when increasing the sparsity ratio, and at higher sparsity (80%), it turns out to be model-dependent.

Table 33: Perplexity achieved by NEURONAL with different calibration data seeds (0, 16, 46) at 60% sparsity.

| Dataset | Model | | | | |
|---|---|---|---|---|---|
| | Phi-2.7B | LLama-1 7B | LLama-2 7B | Mistral-7B | OPT-6.7B |
| WikiText2 | $24.5 \pm 0.6$ | $9.5 \pm 0.1$ | $9.3 \pm 0.0$ | $10.1 \pm 0.1$ | $16.2 \pm 0.1$ |
| C4 | $27.0 \pm 0.2$ | $11.9 \pm 0.1$ | $11.9 \pm 0.0$ | $13.8 \pm 0.1$ | $19.1 \pm 0.1$ |
| PTB | $42.3 \pm 0.7$ | $17.1 \pm 0.3$ | $65.3 \pm 0.8$ | $74.9 \pm 0.2$ | $25.1 \pm 0.1$ |

Table 34: Perplexity achieved by NEURONAL with different calibration data seeds (0, 16, 46) at 80% sparsity.

| Dataset | Model | | | | |
|---|---|---|---|---|---|
| | Phi-2.7B | LLama-1 7B | LLama-2 7B | Mistral-7B | OPT-6.7B |
| WikiText2 | $3654.7 \pm 255.1$ | $382.4 \pm 64.7$ | $247.7 \pm 29.4$ | $216.5 \pm 12.6$ | $1284.9 \pm 482.5$ |
| C4 | $72323.6 \pm 121.7$ | $250.5 \pm 27.1$ | $265.3 \pm 34.1$ | $171.7 \pm 9.9$ | $663.5 \pm 316.3$ |
| PTB | $6014.9 \pm 788.3$ | $624.4 \pm 165.5$ | $1101.9 \pm 94.1$ | $706.1 \pm 6.9$ | $1056.9 \pm 124.5$ |

# E  NeuronAL $\lambda$ Selection at 60% and 80% Sparsity

In the main text, we presented an experiment regarding the ability of NEURONAL to pick the most performing $\lambda$ parameters (in the *block*-only case) at 70% sparsity. Here we include the same analysis at 60% and 80% sparsity. In Fig. 6 and Fig. 7, it is clearly visible how NEURONAL still performs correctly over different sparsity ratios. It is also worth noticing that the calibration data always come from the C4 dataset, and then the results are transferred to the other unknown datasets.

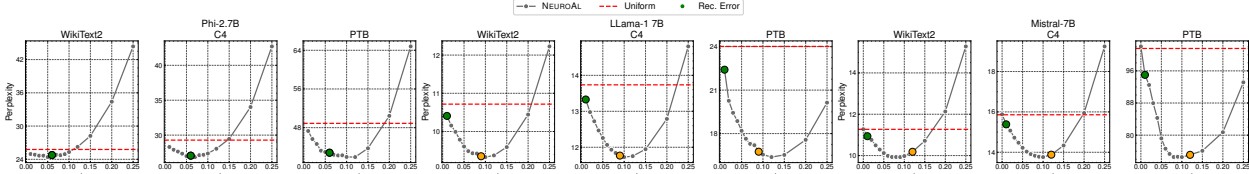

Figure 6: Perplexity over different values of $\lambda$ at 60% sparsity. The orange dot indicates the value selected by NEURONAL.

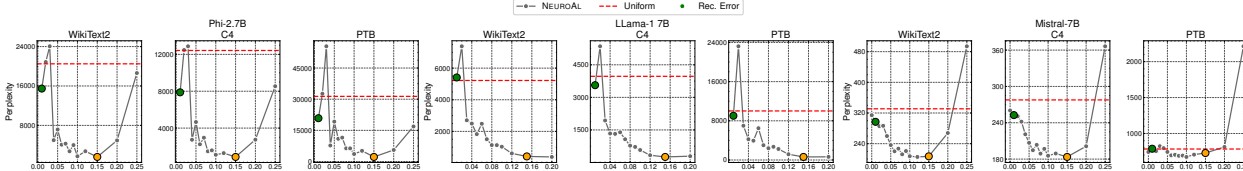

Figure 7: Perplexity over different values of $\lambda$ at 80% sparsity. The orange dot indicates the value selected by NEURONAL.

# F   Reproducibility: $\lambda$ Parameters Selected by NeuronAL

Here we show the $\lambda$ parameters selected by our proposed approach for each model, sparsity ratio, and pruning algorithm tested in this work, aiming to facilitate the reproducibility of our results for the community. Please note that such values are the ones used for each combination of sparsity-pruning algorithm-model that have been extracted from $C_\lambda$ from C4 (using 0 as seed and 8 as size), and then transferred to all the other datasets/tasks. We report the final $\lambda$ values for both the *block* and *row* steps in Table 35 for the first 5 models tested in the main text, and in Table 36 for the LLama 13B models.

Table 35: $\lambda$ parameters selected by NEURONAL (block | row) for each combination of sparsity-pruning algorithm-model. Note that, for SparseGPT, the *row* step is not possible (see the main text for details).

| Sparsity | Top-Up | Phi-2.7B | LLama-1 7B | LLama-2 7B | Mistral-7B | OPT-6.7B |
|---|---|---|---|---|---|---|
| | | | | Model | | |
| 60% | Magnitude | 0.01 \| 0.25 | 0.10 \| 0.20 | 0.20 \| 0.04 | 0.15 \| 0.25 | 0.25 \| 0.25 |
| | Wanda | 0.10 \| 0.25 | 0.09 \| 0.25 | 0.12 \| 0.25 | 0.08 \| 0.00 | 0.01 \| 0.15 |
| | MULTIFLOW | 0.08 \| 0.25 | 0.12 \| 0.25 | 0.12 \| 0.25 | 0.06 \| 0.00 | 0.01 \| 0.06 |
| | SparseGPT | 0.01 \| | 0.02 \| | 0.06 \| | 0.09 \| | 0.02 \| |
| 70% | Magnitude | 0.06 \| 0.25 | 0.20 \| 0.20 | 0.25 \| 0.00 | 0.20 \| 0.25 | 0.15 \| 0.20 |
| | Wanda | 0.12 \| 0.25 | 0.15 \| 0.20 | 0.15 \| 0.25 | 0.15 \| 0.25 | 0.25 \| 0.25 |
| | MULTIFLOW | 0.15 \| 0.25 | 0.15 \| 0.25 | 0.12 \| 0.25 | 0.15 \| 0.01 | 0.25 \| 0.25 |
| | SparseGPT | 0.02 \| | 0.04 \| | 0.08 \| | 0.08 \| | 0.05 \| |
| 80% | Magnitude | 0.01 \| 0.25 | 0.03 \| 0.10 | 0.20 \| 0.05 | 0.20 \| 0.20 | 0.20 \| 0.08 |
| | Wanda | 0.20 \| 0.25 | 0.20 \| 0.25 | 0.20 \| 0.25 | 0.15 \| 0.25 | 0.20 \| 0.20 |
| | MULTIFLOW | 0.12 \| 0.20 | 0.20 \| 0.25 | 0.20 \| 0.20 | 0.15 \| 0.20 | 0.20 \| 0.09 |
| | SparseGPT | 0.06 \| | 0.02 \| | 0.07 \| | 0.08 \| | 0.07 \| |

Table 36: $\lambda$ values selected by NEURONAL on each combination of sparsity-pruning algorithm for LLama 13B (v1 & v2) (block | row).

| Sparsity | Top-Up | LLama-1 13B | LLama-2 13B |
|---|---|---|---|
| | | Model | |
| 60% | Magnitude | 0.10 \| 0.25 | 0.12 \| 0.12 |
| | Wanda | 0.10 \| 0.20 | 0.09 \| 0.25 |
| | MULTIFLOW | 0.15 \| 0.25 | 0.12 \| 0.25 |
| 70% | Magnitude | 0.12 \| 0.25 | 0.25 \| 0.25 |
| | Wanda | 0.15 \| 0.20 | 0.12 \| 0.20 |
| | MULTIFLOW | 0.15 \| 0.20 | 0.15 \| 0.25 |
| 80% | Magnitude | 0.20 \| 0.15 | 0.12 \| 0.05 |
| | Wanda | 0.15 \| 0.20 | 0.15 \| 0.25 |
| | MULTIFLOW | 0.20 \| 0.25 | 0.04 \| 0.25 |

