# OpenReview forum: "Zeroth-Order Adaptive Neuron Alignment Based Pruning without Re-Training"
_TMLR — Accepted by TMLR_

### Review · Reviewer_JQU1 · 2025-08-08

**Summary Of Contributions:**

The paper introduces NeuronAl, a new top-up algorithm for pruning LLMs based on neuron alignment. Neuron alignment compares the dense and sparsified input activations to each projection matrix in a Transformer block on a calibration set $C_\lambda$. The intuition is that these are the real control points of information flow. The method redistributes block-wise and row-wise sparsity based on neuron alignment.

Strengths:
- the method doesn't need hyperparameter tuning, only the size of the calibration set $C_\lambda$ needs to be given
- the method only requires computing the activations (and not the gradients, for example), so it is fast
- interesting insight: the linear sparsity schedule is the most reliable and gives good results
- comprehensive experiments across three language modeling datasets and 7 zero-shot tasks. Three different pruning algorithms and four top-up algorithm baselines are used.
- comprehensive ablation studies about $\lambda$ selection, comparing neuron alignment to using reconstruction error, and sensitivity to calibration data
- the paper measures the inference speed-up after pruning
- the paper is clearly written

Weaknesses:
- NeuronAl is outperformed by AlphaPruning on midsize models (LLama-2 70B), and the zero-shot tasks are not measured
- future work could be discussed

**Audience:**

Yes

**Audience Explanation:**

I believe that the ideas of the paper are novel and interesting, and its topic is very relevant to TMLR.

**Broader Impact Concerns:**

I don't believe the work has ethical implications that require a Broader Impact statement.

**Claims And Evidence:**

Yes

**Claims Explanation:**

The claims are well-supported in general, with extensive evaluations. However, there are some gaps.

My main concern is about the relationship between model size and the performance of NeuronAl. Although the Introduction states that the approach is tested on LLMs from 7B to 70B parameters, most of the results are obtained on the small LLMs (<= 7B parameters). I believe that the only results with Llama-2 70B are in Table 8. NeuronAl - although it is faster - is outperformed by AlphaPruning on the 70B model. I coulnd't find the zero-shot task results for the larger models (I think they are missing).

I believe that the analysis of the runtimes could be expanded. Currently, Table 9 has only the runtime of LLama-7B, but it would be more interesting to see how the runtime scales with the size of the model. Section 5.3.1 mentions that NeuronAl is still about $10\times$ as fast as AlphaPruning on Llama-2 70B, this could be expanded upon. The algorithmic complexity of the methods could be compared.

Lastly, the models examined in the paper (e.g., Llama-1) are dated by LLM standards and have been superseded by newer versions.

**Requested Changes:**

My main requested changes are about making the claims better supported:
- include the results of zero-shot tasks for the larger models, like LLama-2 70B
- provide additional information for the analysis of runtimes

It would be good to include some more recent LLMs if possible.

Small typos:
- OWL is not cited at its first mention.
- "This holds mainly BESA Xu et al. (2024) and DSA Li et al. (2024)."
- "to align with the latest discovering in the literature of structured pruning"

---

> ### Author Response · Authors · 2025-08-25
>
> We thank the reviewer for the valuable feedback. For the sake of clarity, we updated the manuscript, highlighting in green the modifications made concerning the Requested Changes. Below is a short summary of the main modifications.
>
> - In the main text (Section 5.3.1), we included the Zero-Shot tasks results (as average over 7 tasks), while the full results are available in Appendix D.5 for the larger models (Llama 13B, both v1 and v2, and Llama-2 70B). As highlighted in the review, Alphapruning tends to outperform our proposed approach in language modelling over the 70B models. On the other hand, the new results also indicate that NeuronAL outperforms Alphapruning in the majority of the cases on Zero-Shot tasks.
> - As requested, we also included additional results for the pruning runtimes (i.e., the time required to obtain the non-uniform sparsity schedule). The results are in line with the ones previously shown in the paper, see Table 11 (even though we have the numerical values only for the baselines tested in the main text and not for BESA and DSA, for which we took the results directly from the original papers for the LLama 7B model). The additional results, available in Table 10 (Section 5.4.1), further highlight the performance-runtime trade-off of our method: while NeuronAL tends to be 3.5-4x slower than OWL, even when scaling model size, it is drastically faster than AlphaPruning (the strongest baseline), from 6x in the 7B case, up to 20x in the 70B case.
> - We included in the Appendix the results of NeuronAL and the baselines over LLama-3 8B. The results are in line with those shown for the other models reported in the main text. While NeuronAL offers competitive results (although it does not always result in being the best algorithm in all cases) over language modelling datasets, it is by a large margin the most performing algorithm over Zero-Shot tasks on average across seven different tasks.
> - We fixed all the small typos, thanks for spotting them!

---

### Review · Reviewer_3osa · 2025-08-11

**Summary Of Contributions:**

**Summary**:

The paper introduces NeuronAl, which is a top-up algorithm designed to improve the performance of existing pruning methods without retraining. The method uses neuron alignment, i.e. a metric comparing input activations between a dense and pruned model, to guide block-wise and row-wise sparsity redistribution. The proposed method is tested extensively in four LLM families with ten language tasks, showcasing the method’s effectiveness.


**Strengths**:
- The paper is well-written and easy to follow.
- It makes sense to use the activations from both sparse and dense models. Alignment of these metrics is an intuitive idea.
- The experimentation is really thorough with over 300 test cases.
- The empirical results are mostly solid.


**Weaknesses/Requested Changes**:
- One major weakness in my opinion is the lack of theoretical motivation for using the alignment as a metric. While the intuition to use alignment is somewhat clear, a small theoretical explanation would benefit this paper. However, I understand if a full theoretical proof would be too much for a rebuttal, but maybe a couple of paragraphs hinting at the theory would be nice.
- I would also recommend the authors to more explicitly state what are the claims and evidence of the paper in the introduction (for eg. Claims regarding the fact that the method is better/faster,etc.)
- One recommended ablation would be to compare the proposed alignment metric with other potentially good alignment metrics, e.g. cosine-similarity, KL-divergence, etc. This would strengthen the use of the exact formulation used in the paper.

**Minor Weakness**:
- Use citep and citet for citations. This differentiates the in-sentence citations with ones that are not.
- Replacing conference names with references in Figure 1.

**Audience:**

Yes

**Audience Explanation:**

This work will be interesting to the compression community.

**Claims And Evidence:**

Yes

**Claims Explanation:**

The paper claims that the proposed method, using alignment between activations of dense and sparse models, is better than previous approaches, which is demonstrated in the experiments.

**Requested Changes:**

Please see the Weaknesses

---

> ### Author Response · Authors · 2025-08-25
>
> We thank the reviewer for the valuable feedback. For the sake of clarity, we updated the manuscript, highlighting in blue the modifications made concerning the Requested Changes. Below is a short summary of the main modifications.
>
> - We agree that providing a rigorous theoretical motivation for using the proposed neuron alignment metric would be useful. While characterizing analytically the information preserved by alignment is far from being a trivial task, we included in Appendix A a short discussion to clarify the theoretical differences between neuron alignment and the reconstruction error metric that is commonly used in the literature. We do hope that this addition goes in the direction suggested by the reviewer.
> - We revised the introduction to better state the claims and evidence of the paper concerning the better results compared to the selected baselines, the hyperparameter adaptation, and the performance-pruning time trade-off.
> - In Section 5.3.2 we extended the ablation study, including the suggested distance metrics (Cosine Similarity and KL-Divergence) compared to the standard L2-norm. The results shown in Table 13 confirm that in the majority of the cases, the L2-norm formulation outperforms the other tested distance metrics, further validating the effectiveness of the proposed neuron alignment formulation.

---

### Review · Reviewer_KUSh · 2025-08-16

**Summary Of Contributions:**

This paper focuses on unstructured pruning and proposes a new approach, NeuroAL. This method obtains sparse models by maximizing the activations' alignment with respect to their corresponding dense models. It is superior to baseline approaches in that 1. it requires no further training; 2. it runs faster than the baseline approaches. Empirical results of NeuroAL on language modeling and downstream tasks also demonstrate advantages.

While the strengths are already stated above, I have some concerns about the contributions of this work:

1. Across all the evaluations, this sparse model suffers from very significant performance decrease compared to the dense model. This questions the meaning of this line of work in the first place, as the proposed approach does not seem really useful practically given such a  performance loss. I guess maybe this is because the authors perform experiments at relatively large sparsity ratios at 60%, 70%, 80%. I think it is helpful to report results at smaller sparsity ratios to understand better on the sparsity-performance tradeoff.

2. Unstructured pruning is known to provide limited improvements in inference speed, despite achieving seemingly high sparsity levels. This is also reflected in Table 10, which shows only a 1.3x–1.5x speedup at 80% sparsity. Nevertheless, I am curious about which inference engines the authors used for evaluating the dense baseline. Frameworks such as vLLM and sglang have introduced extensive optimizations for dense model inference, while similar optimizations may be either infeasible or highly non-trivial for unstructured-pruned models.

**Audience:**

Yes

**Audience Explanation:**

This paper proposes a new approach to benefit unstructured pruning, which should be relevant to some TMLR's audience.

**Claims And Evidence:**

Yes

**Claims Explanation:**

The authors mainly claim the superiority of the proposed approach over the other unstructured pruning baselines, which I think is accurate.

**Requested Changes:**

1. Report results at more sparsity levels to reveal the full picture of sparsity-performance tradeoff, which can help understand how practical the proposed approach is.
2. Include more experiment details on the inference speed evaluation

---

> ### Author Response · Authors · 2025-08-25
>
> We thank the reviewer for the valuable feedback. For the sake of clarity, we updated the manuscript, highlighting in purple the modifications made concerning the Requested Changes. Below is a short summary of the main modifications.
>
> - We report the results on a broader sparsity range for two different models (LLama 7B and Phi-2) in Appendix D.7. As already shown in previous works [1,2], these results confirm that top-up pruning algorithms provide almost no advantages up to 60% sparsity levels, while they show stronger results at higher sparsity ratios. For this reason, the authors of OWL [1], AlphaPruning [2], and DSnoT [3] mainly test their methods at 70% sparsity levels. We included in Appendix D.7 a clear statement about this sparsity-performance pattern.
> - In Section 5.4.2, we better clarified the inference speed evaluation. For fairness, both dense and sparse models were tested over CPU using the same CPU hardware, from separate ONNX exports of the dense and pruned checkpoints (with identical export settings such as task, opset, and sequence length), using DeepSparse/ONNXRuntime backends.
>
> [1] Yin, Lu, et al. "Outlier weighed layerwise sparsity (OWL) a missing secret sauce for pruning LLMs to high sparsity." Proceedings of the 41st International Conference on Machine Learning. 2024. \
> [2] Lu, Haiquan, et al. "Alphapruning: Using heavy-tailed self regularization theory for improved layer-wise pruning of large language models." Advances in neural information processing systems 37 (2024): 9117-9152. \
> [3] Zhang, Yuxin, et al. "Dynamic Sparse No Training: Training-Free Fine-tuning for Sparse LLMs." The Twelfth International Conference on Learning Representations.

---

### Decision · Action_Editor_Up5Y · 2025-10-23

**Recommendation:** Accept with minor revision

**Additional Comments:**

1. The functions, NeuronAL and GetDist, in Figure 3 are undefined. The 'neuro_al' formula in Equation 2 has different inputs from the function  NeuronAL called in GetBestNeuronAL, and A_D and A_S are not defined consistently.
2. Figure 3, right, the function GetBestNeuronAL does not have the input s but uses s in the middle, while it has S in the input but not using S.
3. The usage of 'top' in Equation 1 is different from Equations 4 and 5.

Minor:
1. Figure 3, left, the 3rd line, there is a missing comma.
2. Table 6 (page 8) is before Table 5 (page 9).

**Audience:**

Yes

**Audience Explanation:**

The improvement in perplexity from this method compared to existing methods does not seem groundbreaking, while the short runtime and freedom from hyperparameter tuning are valuable for practitioners to conduct rapid experiments and determine if this approach can be applied effectively to practical problems.

**Claims And Evidence:**

Yes

**Claims Explanation:**

This paper proposed NeuronAl for LLM pruning without re-training, which achieves good perplexity with low runtime in various models at different sizes and sparsity levels. Their advantage is mainly in 'no need for hyperparameter tuning' and 'faster running using activations instead of gradients'. The authors provided an extensive comparison with other methods and the ablation study.